# DATR: DDI-Aware Therapeutic Structure Reconstruction for Safer Medication Recommendation

## Abstract

Medication recommendation systems play a critical role in clinical decision support, where ensuring both predicting accuracy and safety, particularly drug-drug interaction (DDI) avoidance, is essential. While recent studies have explored drug molecular structures to enhance accuracy, they often overlook the semantic gap between chemical structures and therapeutic outcomes, leading to suboptimal recommendation. Moreover, existing DDI mitigation strategies typically operate in a post-hoc manner, limiting their ability to proactively prevent DDI. In this work, we propose **D**DI-**A**ware **T**herapeutic Structure **R**econstruction (DATR), a novel framework that jointly models drug structures, therapeutic intent, and safety profiles. DATR conditionally encodes drug structures based on ATC-derived therapeutic labels, enabling intent-aware representation learning, and introduces a selectivity potential DDI constraint to proactively reduce interaction risk. Experiments on two real-world datasets and evaluations by clinical experts demonstrate that DATR achieves superior performance in recommendation accuracy and DDI reduction. Code is available at https://anonymous.4open.science/r/DATR-7EA8.

## 1 INTRODUCTION

The rapid digitalization of healthcare has significantly transformed clinical practice, with medication [1] recommendation systems emerging as pivotal tools for enhancing decision-making processes (Dagliati et al., 2021; Garriga et al., 2022). By leveraging computational techniques to analyze electronic health records (EHRs), patient medical histories, and pharmacological profiles, they can assist in selecting effective medication regimens tailored to individual patient needs (Macias et al., 2023). For these systems, their success hinges on two critical factors: **accuracy**, to ensure clinically relevant recommendations with therapeutic efficacy, and **safety**, to prevent adverse outcomes such as drug-drug interactions (DDIs) (Han et al., 2022; Bougiatiotis et al., 2020; Chiang et al., 2020).

For accuracy, capturing the association between drugs and patients' health conditions is of critical importance. To this end, instance-based methods (Zhang et al., 2017) establish associations between drug labels and the patient's current visit record, while longitudinal approaches such as (Shang et al., 2019; Wu et al., 2022) further incorporate historical visits to capture temporal dependencies. Recent advances have increasingly leveraged drug molecular structure information to enrich drug feature representations, achieving improved accuracy (Yang et al., 2021b; 2023; Kuang & Xie, 2024). Though this approach has demonstrated promise, it often assumes a direct correspondence between molecular structures and therapeutic outcomes, overlooking the semantic gap between these two feature spaces (Wen et al., 2023; Xu et al., 2025). In practice, identical structures may mediate divergent therapeutic effects in different clinical contexts, e.g, aspirin's dual use in antithrombosis and analgesia (Fuster & Sweeny, 2011). As a result, such systems may struggle to align structural determinants of efficacy with individualized treatment contexts, thereby compromising recommendation accuracy.

Growing attention has been given to the risk of DDIs in safety aspect. Earlier works achieve preliminary DDI control through implicitly modeling via knowledge graphs (Gong et al., 2021) or reinforcement processing (Zhang et al., 2017). To further improve controllability, recent studies

---

[1]In this paper, "medication" and "drug" are used interchangeably to refer to substances used for the treatment of diseases.

impose explicit DDI-related losses to penalize interacting drugs in the recommendation outcomes (Yang et al., 2021b; 2023; Kuang & Xie, 2024), effectively alleviating the DDI events. However, these methods are inherently post-hoc, treating DDI mitigation as as a separate, corrective step rather than an integral part of the recommendation logic. This decoupling between recommendation and interaction control prevents the model from making clinical optimal decisions considering therapeutic effect and DDIs, introducing a bottleneck in balancing accuracy and safety. Moreover, these penalties rely heavily on specific drug pairs observed during training, which hinders the model's capacity to avoid recommending drugs with high interaction potential, especially when specific interacting pairs have not been explicitly encountered in the training, leading to insufficient control of DDIs.

Considering these factors, in this work we propose **D**DI-**A**ware **T**herapeutic Structure **R**econstruction (DATR) framwork to integrate drug structural information, therapeutic intent and safety profiles into a unified modeling framework to jointly enhance accuracy and safety. To derive therapeautic intent, we first collect drug categorical labels of the Anatomical Therapeutic Chemical (ATC) Classification System (Schellekens et al., 2011). To extract intent-aware structural determinants, we introduce a Therapeutic Structure Reconstruction method (illustrated in Figure 1), which employs conditional probabilistic encoding to map structural information into a latent space based on therapeutic context. Categorical therapeutic structural features of each intent are subsequently sampled from the latent space based on the constructed conditional probabilities. This novel method ensures the precise extraction of therapeutic determinants within drug structures across varying efficacy contexts, thereby establishing more reliable association between drugs and patient-specific conditions.

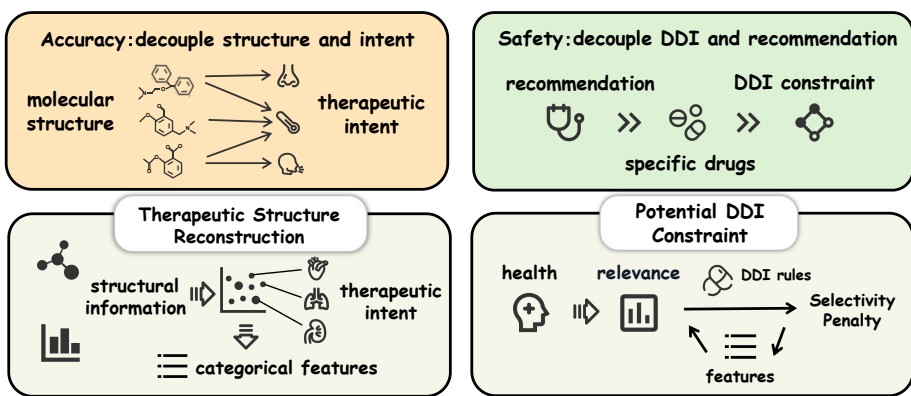

Figure 1: Methods proposed to jointly improve accuracy and safety of medication recommendation

Furthermore, DATR introduces a potential DDI constraint to proactively mitigate adverse interactions while maximizing therapeutic outcomes. As shown in Figure 1, the categorical features derived from reconstruction are utilized to quantify the relevance of each drug category to the health condition of patient. Then we design an asymmetry-inducing selectivity penalty on the joint relevance of interacting drug pairs from prior DDI knowledge, which explicitly constrains potential DDI risks with selective retention of more therapeutically relevant drug. By making the DDI constraint contingent on each drug's contextualized therapeutic relevance, our approach proactively steers the recommendation process towards clinical optimal decisions, thereby achieving a deeper integration of safety and efficacy. Meanwhile, DATR can avoid dependency on specific drug pairs in the training data due to the global consideration of all drug pairs for potential interacting risks.

Our contributions can be summarized as follows:

- We propose a novel framework, DATR, to bridge the semantic gap between chemical structures and clinical outcomes while inherently integrating safety considerations into the recommendation process by jointly modeling drug molecular structures, therapeutic intent, and safety profiles.

- We introduce a therapeutic structure reconstruction method to extract therapeutic structural determinants and design a potential DDI constraint mechanism that imposes a selectivity penalty on joint relevance of interacting drugs to enable proactive DDI avoidance.

- Extensive experiments on two real-world benchmark datasets empirically validate the effectiveness of our approach, which achieves state-of-the-art results in recommendation accuracy and safety.

## 2 RELATED WORKS

**Medication Recommendation.** Existing approaches of medication recommendation can be broadly categorized into instance-based, longitudinal and structure-based methods. Instance-based methods, such as LEAP (Zhang et al., 2017), focus on patient information from the current visit. These approaches often struggle to account for evolving health conditions. Longitudinal approaches address this limitation by leveraging historical records to model temporal dependencies in patient health. For example, GAMENet (Shang et al., 2019) augments memory networks with a DDI graph to enhance both safety and accuracy. COGNet (Wu et al., 2022) selects from the patient's historical prescription records to recommend new medications, while MICRON (Yang et al., 2021a) emphasizes medication change prediction by analyzing differences between consecutive visits. These models improve personalization but lack detailed consideration of drug molecular structures. Models like SafeDrug (Yang et al., 2021b) and MoleRec (Yang et al., 2023) incorporate molecular graph encoders to explicitly model drug structural information and control DDIs. Furthermore, SHAPE (Liu et al., 2023) introduces adaptive mechanisms to handle variable visit lengths and DrugDoctor (Kuang & Xie, 2024) leverages cross-attention for historical influence modeling.

**Deep-learning-based molecular representations.** Deep learning has facilitated the creation of machine-readable continuous representations of molecular structures (Wigh et al., 2022), which were traditionally represented using discrete formats such as SMILES (Weininger, 1988) or InChI (Heller et al., 2013). For example, (Gilmer et al., 2017; Guo et al., 2023; Hamilton et al., 2017) employ GNN to effectively capture the spatial relationships and structural dependencies between atoms and bonds in molecular graphs. (Hou et al., 2022) proposed a bidirectional-LSTM to identify key structural components in the SMILES sequence. Transformer (Vaswani et al., 2017) architectures have also demonstrated strong performance on SMILES-based and graph-based molecular modeling through global self-attention mechanisms (Luong & Singh, 2024; Maziarka et al., 2024). Furthermore, Variational Autoencoders (VAEs) (Kingma, 2013) have seen increasing adoption in molecular representation learning due to their ability to capture smooth and regularized latent spaces, which facilitates downstream tasks such as novel molecules generation and property optimization (Gómez-Bombarelli et al., 2018; Wang et al., 2022; Martinelli, 2022). Motivated by VAEs' potential to extract general features in continuous latent space to support semantic alignment and conditional reconstruction, in this work we design a conditional VAE-style method to integrate molecular structure and therapeutic intent into a unified representation.

## 3 PROBLEM FORMULATION

**Electronic Health Record (EHR).** An EHR is a structured representation of a patient's medical history, encompassing information from multiple clinical visits. For a patient $x$, the EHR is represented as a sequence $\mathcal{V}^{(x)} = [v^{(1)}, v^{(2)}, \ldots, v^{(N_x)}]$, where $N_x$ is the total number of visits for the patient, and $v^{(i)}$ represents the details of the $i$-th visit. Each visit $v^{(i)}$ consists of three main components: $v^{(i)} = [\mathbf{v}_d^{(i)}, \mathbf{v}_p^{(i)}, \mathbf{v}_m^{(i)}]$. Here, $\mathbf{v}_d^{(i)} \in \{0,1\}^{|D|}$ is a multi-hot vector representing the diagnoses from the set $D = \{d_1, d_2, \ldots, d_{|D|}\}$. Similarly, $\mathbf{v}_p^{(i)} \in \{0,1\}^{|P|}$ is a multi-hot vector representing the procedures (e.g., surgeries or therapies) from the set $P = \{p_1, p_2, \ldots, p_{|P|}\}$. Finally, $\mathbf{v}_m^{(i)} \in \{0,1\}^{|M|}$ is a multi-hot vector representing the medications prescribed during the visit, with $M = \{m_1, m_2, \ldots, m_{|M|}\}$ denoting the set of all medications, where a value of 1 indicates that the corresponding medication was prescribed.

**DDI Graph.** DDI graph is represented as a binary symmetric adjacency matrix $\mathbf{A} \in \{0,1\}^{|M| \times |M|}$. Each entry $\mathbf{A}_{ij} = 1$ indicates known harmful interactions between medications $m_i$ and $m_j$.

**Medication Combination Recommendation.** At time step $t$, given the longitudinal diagnosis sequence $\mathbf{v}_d^t = \left[\mathbf{v}_d^{(1)}, \mathbf{v}_d^{(2)}, \ldots, \mathbf{v}_d^{(t)}\right]$, procedure sequence: $\mathbf{v}_p^t = \left[\mathbf{v}_p^{(1)}, \mathbf{v}_p^{(2)}, \ldots, \mathbf{v}_p^{(t)}\right]$ and medication sequence: $\mathbf{v}_m^{t-1} = \left[\mathbf{v}_m^{(1)}, \mathbf{v}_m^{(2)}, \ldots, \mathbf{v}_m^{(t-1)}\right]$, as well as the DDI graph $\mathbf{A}$, our objective is to learn a drug combination recommendation function $f(\cdot)$ that generates a multi-label output $\hat{\mathbf{m}}^{(t)} \in \{0,1\}^{|M|}$. Specifically, $\hat{\mathbf{m}}^{(t)} = f(\mathbf{v}_d^t, \mathbf{v}_p^t, \mathbf{v}_m^{t-1})$.

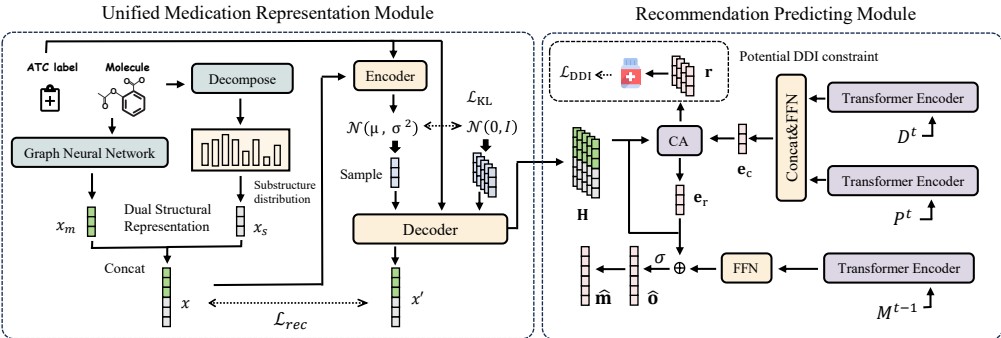

Figure 2: Overview of DATR. The model firstly obtains the unified feature for each medication category by integrating substructure-level and molecule-level therapeutic structural features, $\mathbf{x}_s$ and $\mathbf{x}_m$, through conditional reconstruction based on ATC categorical labels. Then, patient's current health status $\mathbf{e}_c$ is encoded from current visit and interacts with stacked $\mathbf{h}$ of all medication categories to generate recommendations and impose potential DDI constraint. "CA" denotes Cross Attention.

# 4 METHOD

As shown in Figure 2, DATR is composed of: 1) **Unified Medication Representation Module** generating therapeutic structural features for each ATC therapeutic category from both substructure and molecule level. 2) **Recommendation Generating Module** encoding longitudinal health conditions of patients and making predictions base on the condition and medication representations, while integrating potential DDI constraints.

## 4.1 UNIFIED MEDICATION REPRESENTATION MODULE

**Substructure-level Structural Representation.** Considering the varying substructure compositions among medications within the same ATC therapeutic category, we first extract a structural profile at the substructure level. Specifically, we decompose each medication into a set of chemical substructures utilizing *breaking retrosynthetically interesting chemical substructures* (BRICS) (Degen et al., 2008) method. Based on the decomposition results, we construct a substructure probability distribution vector $\mathbf{x_s} \in \mathbb{R}^d$, for each medication, where the $i$-th entry is defined as $\mathbf{x_s}[i] = \frac{f(\mathbf{s}i)}{\sum j=1^d f(\mathbf{s}_j)}$, in which $f(\mathbf{s}_i)$ denotes the frequency of substructure $\mathbf{s}_i$ within the given medication molecule, and $d$ is the total number of distinct substructures across the medications.

**Molecule-level Structural Representation.** While several recent studies have focused primarily on substructure-level representations (Kuang & Xie, 2024; Yang et al., 2023), we argue that modeling molecular structures holistically remains essential. This is especially relevant for biological macromolecule drugs such as insulin whose therapeutic efficacy depends on their overall structural conformation rather than discrete subcomponents (Tanford & Reynolds, 2003; Perrett, 2007; Jones & Thornton, 1996; Petersen & Shulman, 2018).

We employ a GNN (Xu et al., 2018) to encode the molecular graph of each medication. Each molecule is represented as a graph $G = (V, E)$, where $V$ represents atoms (nodes) and $E$ represents bonds (edges). The node features $\mathbf{h}_i \in \mathbb{R}^d$ correspond to atomic properties, and edge features $\mathbf{e}_{ij} \in \mathbb{R}^d$ represent bond types. The node update at layer $l + 1$ is given by:

$$\mathbf{h}_i^{(l+1)} = \text{MLP}((1 + \epsilon^{(l)})\mathbf{h}_i^{(l)} + \sum_{j \in \mathcal{N}(i)} \mathbf{h}_j^{(l)} + \sum_{(i,j) \in E} \mathbf{e}_{ij}), \qquad (1)$$

where $\epsilon^{(l)}$ is a learnable scalar that controls the relative weight of a node's own features in the aggregation, $\mathbf{h}_i^{(l)}$ is the feature vector of node $i$ at layer $l$, and $\mathcal{N}(i)$ denotes the neighbors of node $i$. After $L$ layers of message passing, the final molecule-level representation is obtained via global sum pooling of node features as $\mathbf{x_m} = \sum_{i \in V} \mathbf{h}_i^{(L)} \in \mathbb{R}^d$.

**Therapeutic Structure Reconstruction.** We first obtain dual-level structural representations through $\mathbf{x} = [\mathbf{x}_s, \mathbf{x}_m] \in \mathbb{R}^{2d}$, then we further embed them together with their corresponding therapeutic intent to extract therapeutic structural determinants. To bridge the semantic gap between molecular structures and therapeutic categories, we introduce a latent vector $\mathbf{z}$ to model the key underlying factors that govern how drug structures contribute to specific therapeutic efficacy. The conditional generative process is expressed as:

$$p(\mathbf{x}|\mathbf{y}) = \int p(\mathbf{x}, \mathbf{z}|\mathbf{y})d\mathbf{z} = \int p(\mathbf{x}|\mathbf{z}, \mathbf{y})p(\mathbf{z}|\mathbf{y})d\mathbf{z}, \tag{2}$$

where $\mathbf{y}$ denotes the embeddings of the ATC therapeutic class. A variational distribution $q(\mathbf{z}|\mathbf{x}, \mathbf{y})$ to is introduced to approximate the true posterior $p(\mathbf{z}|\mathbf{x}, \mathbf{y})$. Then we employ the Kullback-Leibler (KL) divergence $\mathrm{KL}(q(\mathbf{z}|\mathbf{x}, \mathbf{y}) \,\|\, p(\mathbf{z}|\mathbf{x}, \mathbf{y}))$ to optimize this approximation, which leads to a tractable variational lower bound on $\log p(\mathbf{x} \mid \mathbf{y})$, given by:

$$\mathcal{L}(\mathbf{x}, \mathbf{y}) = \mathbb{E}_{q(\mathbf{z}|\mathbf{x}, \mathbf{y})}[\log p(\mathbf{x}|\mathbf{z}, \mathbf{y})] - \mathrm{KL}(q(\mathbf{z}|\mathbf{x}, \mathbf{y}) \,\|\, p(\mathbf{z}|\mathbf{y})). \tag{3}$$

This conditional ELBO serves as the reconstruction objective in our framework.

To ensure the continuity of the latent space and the simplicity of computation, we set the conditional prior $p(\mathbf{z}|\mathbf{y})$ to a standard Gaussian distribution $\mathcal{N}(\mathbf{0}, \mathbf{I})$. We parameterize the variational posteriors as Gaussian distributions

$$q_\phi(\mathbf{z} \mid \mathbf{x}, \mathbf{y}) = \mathcal{N}(\mathbf{z}; \mu(\mathbf{x}, \mathbf{y}), \sigma^2(\mathbf{x}, \mathbf{y})), \tag{4}$$

where $\mu(\cdot)$ and $\sigma(\cdot)$ are outputs of learnable neural network encoders. As the KL divergence between Gaussian distributions is analytically tractable, the second term in Equation equation 3 can be computed as follows:

$$\mathcal{L}_{\mathrm{KL}} = -\mathrm{KL}(q(\mathbf{z}|\mathbf{x}, \mathbf{y}) \,\|\, p(\mathbf{z}|\mathbf{y})) = -\frac{1}{2}\sum_{i=1}^{k}\left(1 + \log(\sigma_i^2) - \mu_i^2 - \sigma_i^2\right), \tag{5}$$

in which $k$ denotes the dimension of $\mathbf{z}$. Given the parameterized latent distribution, we can perform sampling to obtain instances of the latent vector $\mathbf{z}$. To enable gradient-based optimization, we apply the reparameterization trick (Kingma, 2013) to obtain $\mathbf{z} = \mu(\mathbf{x}, \mathbf{y}) + \sigma(\mathbf{x}, \mathbf{y}) \cdot \epsilon$, where $\epsilon \sim \mathcal{N}(\mathbf{0}, \mathbf{I})$.

To instantiate $p(\mathbf{x}|\mathbf{z}, \mathbf{y})$, we use a neural network decoder $f(\mathbf{z}, \mathbf{y})$ that predicts the reconstructed $\mathbf{x}' = f(\mathbf{z}, \mathbf{y})$ given instance of $\mathbf{z}$ and $\mathbf{y}$. Maximizing $\log p(\mathbf{x}|\mathbf{z}, \mathbf{y})$ is therefore equivalent to minimizing the mean squared error (MSE) between $\mathbf{x}$ and $\mathbf{x}'$. The first term in equation 3 can be denoted as:

$$\mathcal{L}_{rec} = \mathbb{E}_{q(\mathbf{z}|\mathbf{x}, \mathbf{y})}[\log p(\mathbf{x}|\mathbf{z}, \mathbf{y})] \sim -\mathbb{E}[\|\mathbf{x} - \hat{\mathbf{x}}\|^2] \tag{6}$$

Finally, for each ATC category with its therapeutic label embedding $\mathbf{y}$, we sample latent vector $\mathbf{z}$ from the conditional prior $p(\mathbf{z}|\mathbf{y})$ and obtain the reconstructed therapeutic substructure features $\mathbf{x} \in \mathbb{R}^d$ through learned $f(\mathbf{z}, \mathbf{y})$. Then we stack them as matrix $\mathbf{H} = \left[\mathbf{x}^1; \mathbf{x}^2; \ldots; \mathbf{x}^{|\mathsf{M}|}\right] \in \mathbb{R}^{|M| \times 2d}$.

## 4.2 Recommendation Prediction Module

**Patient Health Condition Encoding.** We utilize three learnable embedding matrices $\mathbf{E}_d$, $\mathbf{E}_p$, and $\mathbf{E}_m$ to encode the diagnosis sequence $\mathbf{v}_d^{(t)}$, the procedure sequence $\mathbf{v}_p^{(t)}$ and the medication procedure sequence $\mathbf{v}_m^{(t-1)}$. These embedded sequences are then passed through three transformer encoders (Vaswani et al., 2017), denoted as $\mathrm{T}(\cdot)$, to capture the dependencies across each visit, resulting in the following encoded representations for the current visit:

$$\mathbf{h}_{\mathrm{d}}^{(t)} = \mathrm{T}(\mathbf{E}_d\mathbf{v}_d^{(t)}), \quad \mathbf{h}_{\mathrm{p}}^{(t)} = \mathrm{T}(\mathbf{E}_p\mathbf{v}_p^{(t)}), \quad \mathbf{h}_{\mathrm{m}}^{(t-1)} = \mathrm{T}(\mathbf{E}_m\mathbf{v}_m^{(t-1)}). \tag{7}$$

The patient's current health condition at time $t$ is obtained by concatenating the encoded diagnosis and procedure representations through a feed-forward network: $\mathbf{e}_c = FFN\left(\left[\mathbf{h}_{\mathrm{d}}^{(t)}, \mathbf{h}_{\mathrm{p}}^{(t)}\right]\right) \in \mathbb{R}^{2d}$. Previous medication usage condition is denoted by $\mathbf{e}_m = FFN(\mathbf{h}_{\mathrm{m}}^{(t-1)}) \in \mathbb{R}^{|M|}$.

**Potential DDI constraint.** We integrate DDI knowledge by first assessing the therapeutic relevance of each medication category $k$ in the context of the patient's current health condition $\mathbf{e}_c$. Specifically, we compute cross-attention weights as a vector $\mathbf{r} \in [0,1]^{|M|}$:

$$\mathbf{r} = \text{softmax}(\frac{(\mathbf{e}_c \mathbf{W}_q)(\mathbf{H} \mathbf{W}_k)^T}{\sqrt{d}}), \tag{8}$$

where $\mathbf{W}_q$ and $\mathbf{W}_k$ are linear transformation matrix. Each element $r_i$ indicating the therapeutic relevance score corresponding to drug category $i$. For two drugs exhibiting high therapeutic relevance but demonstrating adverse interaction, it is critical to reduce their joint relevance to mitigate potential DDI risks while the drug with higher therapeutic relevance in the interacting pair should be prioritized and retained in the regimen to preserve maximal treatment outcomes. To achieve this, we incorporate $\mathbf{r}$ into a global selectivity penalty formally expressed as:

$$\mathcal{L}_{\text{DDI}} = \sum_{i=1}^{|M|} \sum_{j=i+1}^{|M|} \mathbf{A}_{ij} \cdot r_i \cdot r_j \cdot [(1-r_i)^\alpha \sigma(\beta(r_j - r_i)) + (1-r_j)^\alpha \sigma(\beta(r_i - r_j))]. \tag{9}$$

Here, the term $\mathbf{A}_{ij} \cdot r_i \cdot r_j$ penalizes joint relevance of drug pairs $(i,j)$ with known interactions ($\mathbf{A}_{ij} > 0$). The asymmetry-inducing terms $(1-r_i)^\alpha \sigma(\beta(r_j - r_i))$ and $(1-r_j)^\alpha \sigma(\beta(r_i - r_j))$ where $\sigma$ denotes sigmoid activation function, encourage the retention of more therapeutically relevant drug while suppressing its interacting counterpart. The parameters $\alpha$ and $\beta$ control the sharpness and directional sensitivity of this penalty.

**Recommendation Prediction.** We leverage patient-specific and safety-informed relevance scores $\mathbf{r}$ to derive a context vector $\mathbf{e}_\mathbf{r} = \mathbf{r} \cdot (\mathbf{H} \mathbf{W}_\mathbf{v}) \in \mathbb{R}^d$, which summarizes the relevant therapeutic landscape for the patient. The final prediction probability $\hat{\mathbf{o}}$ is obtained by:

$$\hat{\mathbf{o}} = \sigma(\mathbf{e}_r \mathbf{H}^T + \mathbf{e}_m). \tag{10}$$

Following previous work (Shang et al., 2019; Yang et al., 2021b), we treat the final prediction of each medication as an independent task and use the BCE loss for recommendation optimization:

$$\mathcal{L}_{\text{BCE}} = -\sum_{i=1}^{|M|} [m_i \log(\hat{o}_i) + (1 - m_i) \log(1 - \hat{o}_i)]. \tag{11}$$

The model is trained end-to-end by optimizing a total loss function defined as

$$\mathcal{L} = \mathcal{L}_{\text{rec}} + \mathcal{L}_{\text{KL}} + \mathcal{L}_{\text{BCE}} + \gamma \mathcal{L}_{\text{DDI}}, \tag{12}$$

where $\gamma$ is a hyperparameter to regulate the influence of DDI constraint. The multi-label medication combination output $\hat{\mathbf{m}}^{(t)} \in \{0,1\}^{|\mathcal{M}|}$ can be derived from $\hat{\mathbf{o}}$ through thresholding.

# 5 EXPERIMENTS

In this section, we conduct extensive experiments to make a comprehensive evaluation of our proposed method and answer the following four questions: **RQ1:** How does the performance of the proposed DATR compare to that of existing medication recommendation methods? **RQ2:** Does DATR effectively mitigate DDIs while maximizing therapeutic outcomes? **RQ3:** How do the different components of DATR contribute to its performance in terms of both accuracy and safety? **RQ4:** How do the hyperparameters affect the recommendation performance and safety of DATR?

## 5.1 EXPERIMENT SETUP

**Dataset.** We utilized electronic health record data from two real-world EHR datasets, specifically MIMIC-III (Johnson et al., 2016) and MIMIC-IV (Johnson et al., 2023). In line with prior studies (Shang et al., 2019; Yang et al., 2021b), the datasets were processed and randomly split into training, validation, and testing sets with a ratio of 4:1:1. Details of dataset can be found in Appendix C.2.

**Evaluation Metrics.** We use four commonly adopted metrics in medication recommendation (Shang et al., 2019; Yang et al., 2021b; 2023; 2021a): Drug-Drug-Interaction Rate (DDI), Jaccard Similarity Score (Jaccard), F1-score, and Precision-Recall Area Under Curve (PRAUC). DDI is a safety-related

metric that calculates the rate of predicted combinations that involve two or more drugs with a positive relationship in the DDI matrix. The other metrics, Jaccard, F1, and PRAUC, are commonly used to assess the accuracy of recommendation systems in general recommendation literature. Specifically, higher values of Jaccard, PRAUC, and F1 indicate improved accuracy, while a lower DDI value suggests more secure.

**Baselines.** We compare the proposed DATR with the following 10 baseline methods. Instance-based methods: standard logistic regression (LR), LEAP (Zhang et al., 2017). Longitudinal modeling methods: GAMENet (Shang et al., 2019), MICRON (Yang et al., 2021a), COGNet (Wu et al., 2022), SHAPE (Liu et al., 2023), RAREMed (Zhao et al., 2024). Molecular structure informed methods: SafeDrug (Yang et al., 2021b), MoleRec (Yang et al., 2023), DrugDoctor (Kuang & Xie, 2024). Detailed introduction of baselines can be found in Appendix C.5.

**Implementation Details.** The hyperparameters of all baseline models are selected based on their performance on the validation set. For our proposed DATR model, hyperparameters are tuned via grid search to ensure optimal performance. Specifically, we set the embedding dimension of the Transformer encoder to 128, with 4 attention heads and 2 layers. The drug molecular encoder is implemented using a 3-layer Graph Isomorphism Network (GIN) (Xu et al., 2018), with each layer having an embedding dimension of 128. Each drug's molecular graph is represented using node features of dimension 9 and edge features of dimension 3. The transformation functions $\mathbf{q}_\phi$ and $\mathbf{q}_\psi$ are implemented as two-layer biased linear projections. We train the model for 70 epochs. The hyperparameters are set as $\alpha = 1.0$, $\beta = 4$, and $\gamma = 0.1$. Optimization is performed using the AdamW optimizer (Loshchilov, 2017) with a learning rate of 1e-4 and a weight decay of 1e-3. All experiments are conducted on an NVIDIA A100 GPU with 80 GB of memory.

## 5.2 OVERALL PERFORMANCE COMPARISON (RQ1)

Table 1 summarizes the overall performance of all methods. The results highlight distinct trends among the different approaches and underscore the strengths of the proposed DATR method. Methods like LR and LEAP, which focus solely on the current visit's patient status, consistently exhibit the lowest performance across most metrics. Longitudinal-based methods, such as GAMENet, demonstrate improved prediction accuracy compared to the baseline methods, indicating the value of incorporating patient history. RAREMed appears to achieve a lower DDI rate by recommending fewer medications, which might implicitly reduce the likelihood of interactions. Safedrug and MoleRec leverage drug structural information to improve prediction, while introducing explicit loss function to mitigate DDI. SHAPE and DrugDoctor achieve better prediction results by learning visit-level knowledge while DrugDoctor, in particular, stands out as the runner-up in most accuracy metrics for both datasets by integrating molecular structural information.

Table 1: Performance of DATR on MIMIC-III and MIMIC-IV datasets. The best and the runner-up results in each column are highlighted in **bold** and underlined, respectively. Performance metrics are presented as mean with standard deviation in subscript.

| Method | MIMIC-III | | | | MIMIC-IV | | | |
| --- | --- | --- | --- | --- | --- | --- | --- | --- |
| | Jaccard ↑ | PRAUC ↑ | F1 ↑ | DDI ↓ | Jaccard ↑ | PRAUC ↑ | F1 ↑ | DDI ↓ |
| LR | $0.4935_{\pm 0.005}$ | $0.7634_{\pm 0.004}$ | $0.6512_{\pm 0.005}$ | $0.0788_{\pm 0.002}$ | $0.4152_{\pm 0.006}$ | $0.6783_{\pm 0.005}$ | $0.5651_{\pm 0.006}$ | $0.0732_{\pm 0.002}$ |
| LEAP | $0.4521_{\pm 0.007}$ | $0.6581_{\pm 0.006}$ | $0.6152_{\pm 0.007}$ | $0.0720_{\pm 0.003}$ | $0.3909_{\pm 0.008}$ | $0.5542_{\pm 0.007}$ | $0.5439_{\pm 0.008}$ | $0.0550_{\pm 0.002}$ |
| GAMENet | $0.5210_{\pm 0.004}$ | $0.7780_{\pm 0.003}$ | $0.6762_{\pm 0.004}$ | $0.0781_{\pm 0.002}$ | $0.4401_{\pm 0.005}$ | $0.6833_{\pm 0.004}$ | $0.5933_{\pm 0.005}$ | $0.0718_{\pm 0.002}$ |
| COGNet | $0.5109_{\pm 0.005}$ | $0.7665_{\pm 0.004}$ | $0.6615_{\pm 0.005}$ | $0.0737_{\pm 0.002}$ | $0.4313_{\pm 0.006}$ | $0.6712_{\pm 0.005}$ | $0.5850_{\pm 0.006}$ | $0.0866_{\pm 0.003}$ |
| RAREMed | $0.5342_{\pm 0.003}$ | $0.7820_{\pm 0.002}$ | $0.6938_{\pm 0.003}$ | $0.0530_{\pm 0.001}$ | $0.4620_{\pm 0.004}$ | $0.6965_{\pm 0.003}$ | $0.6152_{\pm 0.004}$ | $0.0510_{\pm 0.001}$ |
| MICRON | $0.5119_{\pm 0.004}$ | $0.7690_{\pm 0.004}$ | $0.6676_{\pm 0.004}$ | $0.0610_{\pm 0.002}$ | $0.4495_{\pm 0.006}$ | $0.6753_{\pm 0.005}$ | $0.6033_{\pm 0.006}$ | $0.0502_{\pm 0.002}$ |
| SHAPE | $0.5348_{\pm 0.001}$ | $0.7791_{\pm 0.003}$ | $0.6885_{\pm 0.004}$ | $0.0850_{\pm 0.003}$ | $0.4659_{\pm 0.005}$ | $0.6928_{\pm 0.004}$ | $0.6171_{\pm 0.005}$ | $0.0917_{\pm 0.003}$ |
| SafeDrug | $0.5255_{\pm 0.004}$ | $0.7732_{\pm 0.003}$ | $0.6804_{\pm 0.004}$ | $0.0688_{\pm 0.002}$ | $0.4560_{\pm 0.005}$ | $0.6858_{\pm 0.004}$ | $0.6098_{\pm 0.004}$ | $0.0689_{\pm 0.002}$ |
| MoleRec | $0.5303_{\pm 0.002}$ | $0.7795_{\pm 0.003}$ | $0.6844_{\pm 0.004}$ | $0.0692_{\pm 0.002}$ | $0.4502_{\pm 0.005}$ | $0.6867_{\pm 0.004}$ | $0.6040_{\pm 0.005}$ | $0.0699_{\pm 0.002}$ |
| DrugDoctor | $0.5422_{\pm 0.003}$ | $0.7813_{\pm 0.002}$ | $0.6975_{\pm 0.003}$ | $0.0603_{\pm 0.002}$ | $0.4703_{\pm 0.004}$ | $0.6988_{\pm 0.003}$ | $0.6190_{\pm 0.004}$ | $0.0705_{\pm 0.002}$ |
| DATR | **$0.5506_{\pm 0.003}$** | **$0.7905_{\pm 0.002}$** | **$0.7073_{\pm 0.003}$** | **$0.0366_{\pm 0.002}$** | **$0.4783_{\pm 0.002}$** | **$0.7020_{\pm 0.002}$** | **$0.6216_{\pm 0.003}$** | **$0.0425_{\pm 0.001}$** |

Our proposed DATR method consistently and significantly outperforms all other evaluated methods across both the MIMIC-III and MIMIC-IV datasets. It achieves the highest Jaccard, PRAUC, and F1 scores, indicating superior predictive accuracy in identifying relevant medications. For instance, on MIMIC-III, DATR's Jaccard score is approximately 1.7% higher than the runner-up (DrugDoctor).

Crucially, DATR also achieves the lowest DDI rate, demonstrating its exceptional effectiveness in recommending safer medication combinations. On MIMIC-III, DATR reduces the DDI rate by approximately 39% compared to the runner-up with the lowest DDI (RAREMed), and by over 60% compared to DrugDoctor. This strong performance across both accuracy and safety metrics highlights the effectiveness of DATR's approach, which uniquely integrates drug molecular structural information, therapeutic intent and learned safety profiles.

## 5.3 CASE STUDY (RQ2)

To intuitively illustrate the advantages of DATR in mitigating DDIs while maximizing therapeutic outcomes, we randomly selected four patient visits from the test set and conducted a detailed comparative analysis of recommendation results across four models: GAMENet, SafeDrug, MoleRec, and DATR. As shown in Figure 3, we invited a panel of 20 clinical experts to evaluate the recommended medication lists generated by each model.

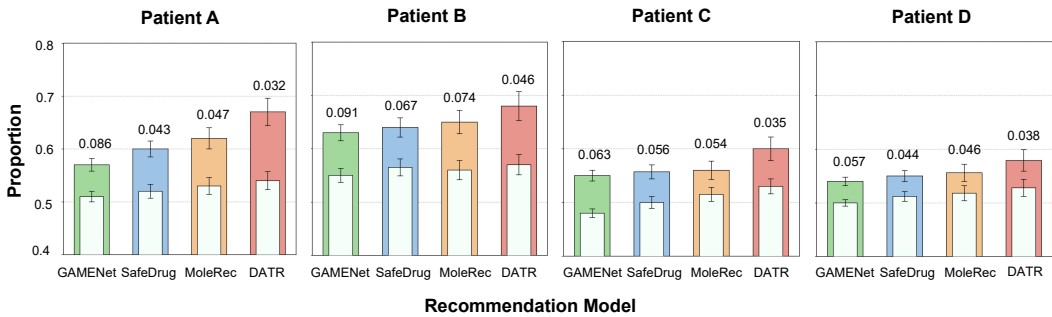

Figure 3: Comparison of four models on four sample patient visits. Inner bar: proportion of drugs overlapping with ground-truth prescriptions; outer bar: proportion of drugs judged effective by clinicians. Numbers above bars indicate DDI rate. DATR achieves the best performance on effectiveness and safety.

From the case study, we observe that DATR consistently achieves a higher proportion of clinically validated drugs, while maintaining a lower DDI rate compared to baseline models. Notably, although certain baseline models, such as SafeDrug and MoleRec, exhibit comparable overlap with ground-truth prescriptions, they often include drug combinations with higher interaction risks or lower expert-judged efficacy. These results underscore DATR's ability to generate recommendations that are not only aligned with historical treatment patterns but also robust to adverse interactions and clinically meaningful, thereby enhancing its potential utility in real-world decision support systems.

During our detailed analysis of the expert evaluations, we noticed a recurring pattern: many drugs that clinicians judged as effective but that were not present in the ground-truth prescriptions were therapeutically interchangeable with medications that were prescribed. This highlights the strength of our proposed therapeutic structure reconstruction method in capturing treatment semantics by modeling drug structure through therapeutic context. Looking ahead, explicitly incorporating the notion of efficacy equivalence into future medication recommendation frameworks may further enhance clinical applicability in scenarios such as drug shortages, patient-specific contraindications, or treatment optimization. Detailed analysis can be found in Appendix E.2.

## 5.4 ABLATION STUDY (RQ3)

To verify the effectiveness of each component of DATR, we design several ablation models. **"w/o $x_m$"** removes the molecule-level structural representation in therapeutic structure reconstruction. **"w/o $x_s$"** removes the substructure-level structural representation in therapeutic structure reconstruction. **"w/o $e_m$"** removes the previous medication usage condition in the recommendation predicting process. To further demonstrate the benefit of therapeutic structure reconstruction, in **"R→T"** we substituted it with a standard Transformer and a pooling layer, and compared the results.

Table 2 presents the performance of the different variants of DATR. Removing either the molecular-level representation $h_m$ or the substructure-level representation $h_s$ leads to a noticeable decline in

recommendation accuracy, underscoring the necessity of capturing drug molecular information from both holistic and fragment-based perspectives. Among all ablations, the most significant performance degradation occurs in the **"R→T"** setting, demonstrating the effectiveness of the proposed therapeutic structure reconstruction in integrating structural features with therapeutic intent.

Furthermore, omitting historical medication embeddings ($e_m$) also results in a reduction in accuracy, suggesting that previous medication usage provides valuable contextual signals for current drug recommendation. In terms of safety, all ablated variants exhibit an increased DDI rate compared to the full model, which emphasizes the importance of unified drug modeling. Nevertheless, the DDI rates of all ablated variants remain relatively low, which empirically illustrates the robustness of our proposed potential DDI constraint mechanism.

Table 2: Ablation study on MIMIC-III dataset.

| Model | Jaccard | PRAUC | F1 | DDI |
|---|---|---|---|---|
| w/o $h_m$ | $0.5267_{\pm 0.001}$ | $0.7811_{\pm 0.003}$ | $0.6931_{\pm 0.001}$ | $0.0426_{\pm 0.001}$ |
| w/o $h_s$ | $0.5312_{\pm 0.001}$ | $0.7828_{\pm 0.003}$ | $0.6992_{\pm 0.001}$ | $0.0407_{\pm 0.001}$ |
| w/o $e_m$ | $0.5319_{\pm 0.003}$ | $0.7825_{\pm 0.002}$ | $0.6983_{\pm 0.002}$ | $0.0453_{\pm 0.001}$ |
| R→T | $0.5117_{\pm 0.002}$ | $0.7528_{\pm 0.001}$ | $0.6702_{\pm 0.001}$ | $0.0463_{\pm 0.001}$ |
| DATR | $\mathbf{0.5506}_{\pm \mathbf{0.003}}$ | $\mathbf{0.7905}_{\pm \mathbf{0.002}}$ | $\mathbf{0.7073}_{\pm \mathbf{0.003}}$ | $\mathbf{0.0366}_{\pm \mathbf{0.002}}$ |

### 5.5 HYPERPARAMETER STUDY (RQ4)

We conducted a dedicated study to meticulously investigate the influence of hyperparameters on the performance of DATR on the MIMIC-III. Specifically, we considered four key hyperparameters: the sharpness exponent $\alpha$ and directional sensitivity coefficient $\beta$ of the DDI constraint, the DDI loss weight ($\gamma$) and the number of training epochs $\#Epochs$.

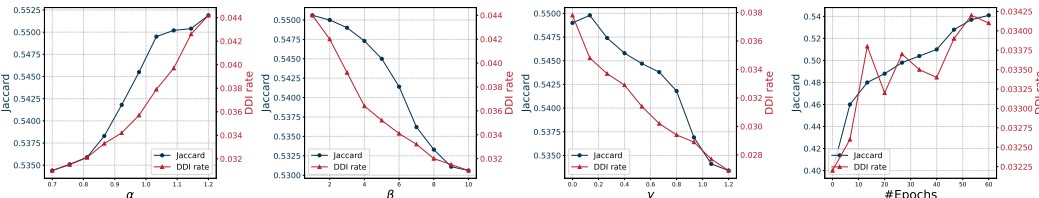

Figure 4: Hyperparameter effects on model performance.

Figure 4 illustrates the impact of different hyperparameters on the model's recommendation accuracy and safety. Increasing $\alpha$ attenuates the penalization applied by the DDI constraint, enhancing accuracy but concurrently lessening the rigor of DDI mitigation. Amplifying $\beta$ heightens the DDI constraint's sensitivity to the varying relevance of interacting drugs, favoring DDI avoidance but diminishing accuracy. Both Jaccard index and DDI rate show a declining trend as $\gamma$ increases. Notably, a modest $\gamma$ can benefit recommendation accuracy, potentially reflecting physicians' consideration of DDIs in real-world prescriptions. During the training process, the DDI rate fluctuates upwards as recommendation accuracy increases, reflecting the influence of DDIs present inherently in the dataset. Nevertheless, the overall DDI rate remains low, demonstrating the advantage of our global consideration of potential DDI.

## 6 CONCLUSION

In this paper, we tackled the critical challenge of simultaneously improving both the effectiveness and safety of medication recommendation systems. We proposed DATR, a novel framework that seamlessly integrates drug molecular structures, therapeutic intent derived from the ATC system, and DDI safety profiles into a unified modeling paradigm. Extensive experiments on two real-world EHR datasets demonstrate that DATR consistently outperforms state-of-the-art baselines. It not only achieves higher accuracy in recommending clinically effective medication combinations but also significantly reduces the incidence of potential drug–drug interactions, offering a promising step toward safer and more reliable clinical decision support.

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

## A  NOTATION

Table 3: Summary of main notations.

| Symbol | Description | Dimension |
|---|---|---|
| $x$ | Index of a patient | $-$ |
| $\mathcal{V}^{(x)}$ | EHR sequence of patient $x$ | $[v^{(1)}, \ldots, v^{(N_x)}]$ |
| $N_x$ | Number of visits of patient $x$ | $\mathbb{N}$ |
| $v^{(i)}$ | $i$-th visit record | $-$ |
| $\mathbf{v}_d^{(i)}$ | Diagnosis multi-hot vector at visit $i$ | $\{0,1\}^{|D|}$ |
| $\mathbf{v}_p^{(i)}$ | Procedure multi-hot vector at visit $i$ | $\{0,1\}^{|P|}$ |
| $\mathbf{v}_m^{(i)}$ | Medication multi-hot vector at visit $i$ | $\{0,1\}^{|M|}$ |
| $D, P, M$ | Sets of diagnoses, procedures, medications | $|D|, |P|, |M|$ |
| $t$ | Current time step / visit index | $\mathbb{N}$ |
| $\mathbf{v}_d^t$ | Diagnosis sequence up to visit $t$ | $[\mathbf{v}_d^{(1)}, \ldots, \mathbf{v}_d^{(t)}]$ |
| $\mathbf{v}_p^t$ | Procedure sequence up to visit $t$ | $[\mathbf{v}_p^{(1)}, \ldots, \mathbf{v}_p^{(t)}]$ |
| $\mathbf{v}_m^{t-1}$ | Medication sequence up to visit $t-1$ | $[\mathbf{v}_m^{(1)}, \ldots, \mathbf{v}_m^{(t-1)}]$ |
| $\mathbf{A}$ | DDI adjacency matrix | $\{0,1\}^{|M| \times |M|}$ |
| $\mathbf{A}_{ij}$ | Indicator of interaction between $m_i, m_j$ | $\{0,1\}$ |
| $f(\cdot)$ | Medication recommendation function | $\{0,1\}^{|M|}$ |
| $\hat{\mathbf{m}}^{(t)}$ | Predicted medication combination at visit $t$ | $\{0,1\}^{|M|}$ |
| $\mathbf{x}_s$ | Substructure-level structural distribution | $\mathbb{R}^d$ |
| $\mathbf{x}_m$ | Molecule-level structural representation | $\mathbb{R}^d$ |
| $\mathbf{x}$ | Concatenated structural feature $[\mathbf{x}_s, \mathbf{x}_m]$ | $\mathbb{R}^{2d}$ |
| $d$ | Dimension of structural feature vectors | $\mathbb{N}$ |
| $G = (V, E)$ | Molecular graph (atoms and bonds) | $-$ |
| $\mathbf{h}_i^{(l)}$ | Node (atom) feature at layer $l$ | $\mathbb{R}^d$ |
| $\mathbf{e}_{ij}$ | Edge (bond) feature | $\mathbb{R}^d$ |
| $L$ | Number of GNN layers | $\mathbb{N}$ |
| $\mathbf{y}$ | ATC therapeutic label embedding | $\mathbb{R}^{d_y}$ |
| $\mathbf{z}$ | Latent variable for therapeutic structure | $\mathbb{R}^k$ |
| $k$ | Dimension of latent variable $\mathbf{z}$ | $\mathbb{N}$ |
| $q_\phi(\mathbf{z} \mid \mathbf{x}, \mathbf{y})$ | Variational posterior | $\mathcal{N}(\mu, \sigma^2)$ |
| $p(\mathbf{z} \mid \mathbf{y})$ | Conditional prior of $\mathbf{z}$ | $\mathcal{N}(\mathbf{0}, \mathbf{I})$ |
| $f(\mathbf{z}, \mathbf{y})$ | Decoder for structure reconstruction | $\mathbb{R}^{2d}$ |
| $\hat{\mathbf{x}}$ | Reconstructed structural feature | $\mathbb{R}^{2d}$ |
| $\mathcal{L}_{\text{rec}}$ | Reconstruction loss | $\mathbb{R}$ |
| $\mathcal{L}_{\text{KL}}$ | KL divergence term | $\mathbb{R}$ |
| $\mathbf{E}_d, \mathbf{E}_p, \mathbf{E}_m$ | Embedding matrices for codes | $\mathbb{R}^{(|D|, |P|, |M|) \times d}$ |
| $\mathbf{h}_d^{(t)}$ | Encoded diagnosis feature at visit $t$ | $\mathbb{R}^d$ |
| $\mathbf{h}_p^{(t)}$ | Encoded procedure feature at visit $t$ | $\mathbb{R}^d$ |
| $\mathbf{h}_m^{(t-1)}$ | Encoded medication feature up to $t-1$ | $\mathbb{R}^d$ |
| $\mathbf{e}_c$ | Current health condition representation | $\mathbb{R}^{2d}$ |
| $\mathbf{e}_m$ | Historical medication usage representation | $\mathbb{R}^{|M|}$ |
| $\mathbf{H}$ | Stacked therapeutic structural features | $\mathbb{R}^{|M| \times 2d}$ |
| $\mathbf{W}_q, \mathbf{W}_k, \mathbf{W}_v$ | Projection matrices in cross-attention | Appropriate sizes |
| $\mathbf{r}$ | Therapeutic relevance scores | $[0,1]^{|M|}$ |
| $r_i$ | Relevance of medication $i$ to current condition | $[0,1]$ |
| $\mathbf{e}_r$ | Context vector aggregated by $\mathbf{r}$ | $\mathbb{R}^d$ |
| $\hat{\mathbf{o}}$ | Predicted medication probabilities | $[0,1]^{|M|}$ |
| $\alpha$ | Sharpness exponent in selectivity penalty | $\mathbb{R}_+$ |
| $\beta$ | Directional sensitivity coefficient | $\mathbb{R}_+$ |
| $\gamma$ | Weight of DDI loss | $\mathbb{R}_+$ |
| $\mathcal{L}_{\text{BCE}}$ | Binary cross-entropy loss | $\mathbb{R}$ |

# B   DISCLOSURE ON THE USE OF LARGE LANGUAGE MODELS

Throughout the research and writing process for this paper, we utilized a Large Language Model (LLM) to assist with specific technical and linguistic tasks. We provide this statement to transparently detail its role.

- **Manuscript Writing and Polishing**: The LLM served as an advanced grammar and style checker. We used it to refine sentence structures, enhance the flow and clarity of our arguments, and ensure consistent terminology. The intellectual contribution, including the formulation of the problem, the proposed methodology, and the interpretation of our findings, originates entirely from the authors.

- **LaTeX and Table Formatting**: The LLM was employed as a technical tool to generate LaTeX code for the presentation of our results, particularly for typesetting complex tables. This streamlined the formatting process but did not influence the content or design of the tables themselves.

- **Experimental Code Implementation:** During the implementation of our experiments, the LLM acted as a coding assistant. Its role was to generate boilerplate code for standard tasks (e.g., file I/O, argument parsing) and to provide syntactical guidance for specific Python libraries. All core algorithmic logic and experimental designs were developed by the authors. Furthermore, any code snippet suggested by the LLM was rigorously tested, and often modified, by the authors before integration into the final codebase.

All LLM-generated outputs, both text and code, were carefully reviewed, verified, and edited by the authors to ensure their accuracy and appropriateness. The conceptualization of the research, the design of the proposed model, the experimental setup, and the interpretation of the results were performed solely by the human authors, who bear full responsibility for the content of this work.

# C   DETAILS ON DATA AND EXPERIMENT SETUP

## C.1   INTRODUCTION OF ATC SYSTEM

The Anatomical Therapeutic Chemical (ATC) Classification System is a widely recognized international standard for classifying drugs based on their organ or system of action and their chemical, pharmacological, and therapeutic properties (Schellekens et al., 2011; Garbe et al., 1993). It is maintained by the World Health Organization (WHO) Collaborating Centre for Drug Statistics Methodology (WHOCC). The ATC system provides a hierarchical structure with five distinct levels, offering increasingly specific classifications from broad anatomical groups to individual chemical substances. Each ATC code uniquely identifies a drug or group of drugs within this hierarchical framework. The increasing specificity of the levels allows for a detailed categorization that reflects both the therapeutic application and, at lower levels, the chemical nature of the drug. Table 4 summarizes the distribution of codes and pharmaceuticals across these levels, along with their semantic meanings.

In this work, we leverage the ATC Classification System, specifically focusing on ATC4 codes. The reason for choosing ATC4 is its ability to bridge the gap between therapeutic intent and chemical structure. While higher levels (ATC1-3) primarily focus on anatomical and therapeutic groupings, Level 4 begins to incorporate chemical subgrouping (e.g., A01AA: Fluorine for dental prophylaxis, reflecting a chemical element used for a specific therapeutic purpose). This level provides a good balance, allowing us to align the semantic categories of drug use (therapeutic intent) with a representation that has closer ties to the

Table 4: Distribution of ATC Levels by Codes and Pharmaceuticals. Codes, Pharma., and Semantic respectively represent the number of codes at each level, the number of pharmaceuticals, and their corresponding meanings.

| Level | Codes | Pharma. | Semantic |
|---|---|---|---|
| Level 1 | 14 | 14 | Anatomical group |
| Level 2 | 94 | 94 | Therapeutic group |
| Level 3 | 267 | 262 | Therapeutic subgroup |
| Level 4 | 889 | 819 | Chemical subgroup |
| Level 5 | 5067 | 4363 | Chemical substance |

underlying chemical structures compared to higher ATC levels, which are purely based on therapeutic or anatomical classifications. By utilizing ATC4, we can better capture the therapeutic determinants embedded within drug structures in an intent-aware manner, which is crucial for the Conditional

Therapeutic Structure Reconstruction proposed in our DATR framework (as discussed in Section 4). This allows us to address the semantic gap between chemical structures and therapeutic outcomes more effectively.

## C.2  DETAILS ON DATASETS

We conduct experiments on two widely used real-world electronic health record (EHR) datasets, MIMIC-III (Johnson et al., 2016) and MIMIC-IV (Johnson et al., 2023). Both datasets contain de-identified longitudinal hospitalization records collected from Beth Israel Deaconess Medical Center, enabling the construction of patient-level clinical trajectories for medication recommendation.

**Data preprocessing.**  For each dataset, we extract chronological sequences of patient visits, where each visit consists of: (i) *diagnoses*, recorded as ICD-9 codes in MIMIC-III and ICD-10 codes in MIMIC-IV; (ii) *procedures*, recorded as ICD-9-CM for MIMIC-III and ICD-10-PCS for MIMIC-IV; and (iii) *prescribed medications*. Following standard practice, all medications are mapped to their corresponding ATC codes through publicly available mapping resources, enabling a unified therapeutic categorization across datasets.

**Patient-level splitting.**  To avoid information leakage across different visits of the same patient and to ensure realistic model evaluation, we adopt a *patient-level* data split. Specifically, patients are randomly partitioned into training, validation, and test sets with a ratio of 4:1:1, and all visits from a given patient appear exclusively in one split. This ensures that the model is evaluated on entirely unseen patients, which better reflects the intended clinical deployment scenario and aligns with established practices in EHR-based predictive modeling.

**Dataset statistics.**  Table 5 summarizes the key statistics of the processed datasets, including the total number of patients and visits, vocabulary sizes of diagnoses, procedures, and medications, and the average number of events per visit. Notably, MIMIC-III and MIMIC-IV differ substantially in coding systems (ICD-9 vs. ICD-10) and data sparsity, providing a natural testbed for evaluating the robustness and generalization capability of our proposed framework.

Table 5: Statistics of processed data.

| Item | MIMIC-III | MIMIC-IV |
|---|---|---|
| # of visits / # of patients | 14949/6344 | 19461/7567 |
| dis. / proc. space size | 1959/1440 | 3973/1338 |
| med. space size | 112/141 | 212/302 |
| avg. / max # of visits | 4.92/29 | 7.28/42 |
| avg. / max # of diag. | 13.79/39 | 13.39/39 |
| avg. / max # of proc. | 4.40/28 | 2.57/28 |
| avg. / max # of med. | 26.23/63 | 13.31/70 |

These comprehensive statistics highlight the heterogeneity and complexity of EHR data, underscoring the importance of models capable of generalizing across diverse clinical settings and coding systems.

## C.3  THE DETAILED FEATURES FOR ATOMS, BONDS AND MOLECULAR GLOBAL

Table 6: Overview of atom (node) and bond (edge) features.

| Atomic Features (V) | Bond Features (E) |
|---|---|
| Atomic Number | Bond Type |
| Chirality | Bond Stereo |
| Degree | Conjugation |
| Formal Charge | – |
| Number of Hydrogens | – |
| Radical Electrons | – |
| Hybridization | – |
| Aromaticity | – |
| Ring Membership | – |

A comprehensive overview of the selected atom and bond input features is presented in Table 6. The initial step involves the conversion of the SMILES string into a graph structure using the RDKit

package. This package is employed not only for constructing molecular graphs but also for computing atomic and bond-level features, which serve as critical inputs for subsequent modeling.

## C.4 DETAILS ON EVALUATION METRICS

For a given patient visit, let $M$ denote the complete set of possible medications in the formulary. The ground truth set of prescribed medications is represented by a binary vector $\mathbf{y} \in \{0, 1\}^{|M|}$, where $\mathbf{y}_i = 1$ if medication $i \in M$ was prescribed, and $\mathbf{y}_i = 0$ otherwise. Similarly, the set of medications recommended by the model is represented by a binary vector $\hat{\mathbf{y}} \in \{0, 1\}^{|M|}$. Jaccard, F1 and PRAUC are calculated as follows:

$$\text{Jaccard} = \frac{\{i : \mathbf{y}_i = 1\} \cap \{j : \hat{\mathbf{y}}_j = 1\}}{\{i : \mathbf{y}_i = 1\} \cup \{j : \hat{\mathbf{y}}_i = 1\}}, \tag{13}$$

$$F_1 = \frac{2R \times P}{R + P}, \tag{14}$$

where the recall and precision are formulated as

$$R = \frac{\{i : \mathbf{y}_i = 1\} \cap \{j : \hat{\mathbf{y}}_j = 1\}}{\{i : \mathbf{y}_i = 1\}}, \quad P = \frac{\{i : \mathbf{y}_i = 1\} \cap \{j : \hat{\mathbf{y}}_j = 1\}}{\{j : \hat{\mathbf{y}}_j = 1\}}. \tag{15}$$

$$\text{PRAUC} = \sum_{k=1}^{|M|} P_k(R_k - R_{k-1}), \tag{16}$$

For DDI, we calculate DDI rate as follows:

$$\text{DDI} = \frac{\sum_{l,k \in \{i:\hat{\mathbf{y}}_i=1\}} A_{lk}}{\sum_{l,k \in \{i:\hat{\mathbf{y}}_i=1\}} 1}, \tag{17}$$

where A represents DDI graph define in section 3.

## C.5 DETAILS ON BASELINES

To comprehensively evaluate our proposed method, we compare it with a variety of representative baseline models as follows:

**LR (Logistic Regression):** A classical linear model that independently predicts medications based on the current visit's features, without considering temporal dependencies or inter-visit information.

**LEAP** (Zhang et al., 2017): An LSTM-based sequence modeling approach that encodes each visit as a temporal instance and predicts the next medication set. It captures sequential patterns within the patient's historical medical records but does not explicitly model drug-drug interactions or molecular features.

**GAMENet** Shang et al. (2019): Combines Graph Convolutional Networks (GCNs) and memory networks to jointly learn from Electronic Health Records (EHRs) and a Drug-Drug Interaction (DDI) graph. It constructs a dynamic memory bank of historical visits to support accurate and safe drug recommendation.

**MICRON** (Yang et al., 2021a): Proposes a conditional recurrent residual network that captures inter-visit dynamics and variations in drug usage patterns. It models both temporal continuity and abrupt changes in medication behaviors across visits.

**COGNet** (Wu et al., 2022): Reformulates medication prediction as a sequence generation task. It incorporates a copy-or-predict mechanism to selectively replicate medications from past visits or generate new drugs based on current health status.

**SHAPE** (Liu et al., 2023): Introduces a lightweight intra-visit encoder to effectively model the relationships among medical events within a visit. It generates expressive visit-level representations to enhance the downstream prediction of medications.

**RAREMed** (Zhao et al., 2024): Utilizes two self-supervised pretraining tasks to learn visit-aware patient embeddings. It focuses on capturing individual-specific medication needs and complex interrelations among clinical codes, which improves model generalization in low-resource scenarios.

**SafeDrug** (Yang et al., 2021b): Integrates pharmacological knowledge by incorporating molecular structure embeddings into the recommendation framework. It jointly optimizes for therapeutic effectiveness and safety by penalizing adverse drug interactions during training.

**MoleRec** (Yang et al., 2023): Enhances drug recommendation accuracy by aligning patients' health conditions with relevant molecular substructures. It employs hierarchical attention to identify and leverage key molecular fragments related to a patient's clinical context.

**DrugDoctor** (Kuang & Xie, 2024): Models the causal impact of historical prescriptions on patient outcomes using a cross-attention mechanism. It accounts for both temporal treatment effects and structural similarities among drugs to inform medication selection.

Overall, the existing medication recommendation methods can be broadly categorized into three methodological families, each with distinct advantages and limitations. Instance-based approaches (e.g., LR) rely solely on the information contained in the current visit and thus offer computational simplicity and strong performance when intra-visit features dominate; however, they fail to capture longitudinal treatment patterns. Longitudinal sequence-based methods (e.g. GAMENet, MICRON, COGNet, RAREMed) explicitly incorporate temporal dependencies across visits, enabling more accurate modeling of patient trajectories, though they often lack fine-grained pharmacological knowledge and may struggle with safety considerations. Molecular-structure-informed models (e.g., SafeDrug, MoleRec, DrugDoctor) integrate structural or chemical information to strengthen pharmacological reasoning and improve safety, yet they typically treat molecular structure and therapeutic intent as separate, unaligned modalities. These methodological distinctions highlight the need for a unified framework capable of jointly modeling temporal patterns, structural information, and therapeutic semantics, which is an objective that DATR is designed to address.

# D DETAILS ON METHOD

## D.1 PROOF

We derive the conditional lower bound in our conditional reconstruction module. Let $q(\mathbf{z} \mid \mathbf{x}, \mathbf{y})$ be a variational distribution that approximates the true posterior $p(\mathbf{z} \mid \mathbf{x}, \mathbf{y})$. Consider the Kullback–Leibler (KL) divergence between these two distributions:

$$\mathrm{KL}\big(q(\mathbf{z} \mid \mathbf{x}, \mathbf{y}) \,\|\, p(\mathbf{z} \mid \mathbf{x}, \mathbf{y})\big) = \mathbb{E}_{q(\mathbf{z}|\mathbf{x},\mathbf{y})}\left[\log \frac{q(\mathbf{z} \mid \mathbf{x}, \mathbf{y})}{p(\mathbf{z} \mid \mathbf{x}, \mathbf{y})}\right]. \tag{18}$$

Using Bayes' rule, the true posterior can be expressed as

$$p(\mathbf{z} \mid \mathbf{x}, \mathbf{y}) = \frac{p(\mathbf{x}, \mathbf{z} \mid \mathbf{y})}{p(\mathbf{x} \mid \mathbf{y})}, \tag{19}$$

and hence

$$\log p(\mathbf{z} \mid \mathbf{x}, \mathbf{y}) = \log p(\mathbf{x}, \mathbf{z} \mid \mathbf{y}) - \log p(\mathbf{x} \mid \mathbf{y}). \tag{20}$$

Substituting this into the KL divergence yields

$$\mathrm{KL}\big(q(\mathbf{z} \mid \mathbf{x}, \mathbf{y}) \,\|\, p(\mathbf{z} \mid \mathbf{x}, \mathbf{y})\big) = \mathbb{E}_{q(\mathbf{z}|\mathbf{x},\mathbf{y})}\left[\log q(\mathbf{z} \mid \mathbf{x}, \mathbf{y}) - \log p(\mathbf{z} \mid \mathbf{x}, \mathbf{y})\right] \tag{21}$$

$$= \mathbb{E}_{q(\mathbf{z}|\mathbf{x},\mathbf{y})}\left[\log q(\mathbf{z} \mid \mathbf{x}, \mathbf{y}) - \log p(\mathbf{x}, \mathbf{z} \mid \mathbf{y}) + \log p(\mathbf{x} \mid \mathbf{y})\right]. \tag{22}$$

Note that $\log p(\mathbf{x} \mid \mathbf{y})$ does not depend on $\mathbf{z}$ and can thus be taken outside the expectation:

$$\mathrm{KL}\big(q(\mathbf{z} \mid \mathbf{x}, \mathbf{y}) \,\|\, p(\mathbf{z} \mid \mathbf{x}, \mathbf{y})\big) = -\mathbb{E}_{q(\mathbf{z}|\mathbf{x},\mathbf{y})}\big[\log p(\mathbf{x}, \mathbf{z} \mid \mathbf{y})\big] + \mathbb{E}_{q(\mathbf{z}|\mathbf{x},\mathbf{y})}\big[\log q(\mathbf{z} \mid \mathbf{x}, \mathbf{y})\big] + \log p(\mathbf{x} \mid \mathbf{y}). \tag{23}$$

Rearranging terms gives

$$\log p(\mathbf{x} \mid \mathbf{y}) = \mathrm{KL}\big(q(\mathbf{z} \mid \mathbf{x}, \mathbf{y}) \,\|\, p(\mathbf{z} \mid \mathbf{x}, \mathbf{y})\big) + \mathbb{E}_{q(\mathbf{z}|\mathbf{x},\mathbf{y})}\big[\log p(\mathbf{x}, \mathbf{z} \mid \mathbf{y})\big] - \mathbb{E}_{q(\mathbf{z}|\mathbf{x},\mathbf{y})}\big[\log q(\mathbf{z} \mid \mathbf{x}, \mathbf{y})\big]. \tag{24}$$

Using the factorization $p(\mathbf{x}, \mathbf{z} \mid \mathbf{y}) = p(\mathbf{x} \mid \mathbf{z}, \mathbf{y})\, p(\mathbf{z} \mid \mathbf{y})$, we obtain

$$\log p(\mathbf{x} \mid \mathbf{y}) = \mathrm{KL}\big(q(\mathbf{z} \mid \mathbf{x}, \mathbf{y}) \,\|\, p(\mathbf{z} \mid \mathbf{x}, \mathbf{y})\big) + \mathbb{E}_{q(\mathbf{z}\mid\mathbf{x},\mathbf{y})}\big[\log p(\mathbf{x} \mid \mathbf{z}, \mathbf{y})\big] - \mathrm{KL}\big(q(\mathbf{z} \mid \mathbf{x}, \mathbf{y}) \,\|\, p(\mathbf{z} \mid \mathbf{y})\big). \tag{25}$$

We define the conditional evidence lower bound (ELBO) as

$$\mathcal{L}(\mathbf{x}, \mathbf{y}) = \mathbb{E}_{q(\mathbf{z}\mid\mathbf{x},\mathbf{y})}\big[\log p(\mathbf{x} \mid \mathbf{z}, \mathbf{y})\big] - \mathrm{KL}\big(q(\mathbf{z} \mid \mathbf{x}, \mathbf{y}) \,\|\, p(\mathbf{z} \mid \mathbf{y})\big), \tag{26}$$

so that the identity

$$\log p(\mathbf{x} \mid \mathbf{y}) = \mathcal{L}(\mathbf{x}, \mathbf{y}) + \mathrm{KL}\big(q(\mathbf{z} \mid \mathbf{x}, \mathbf{y}) \,\|\, p(\mathbf{z} \mid \mathbf{x}, \mathbf{y})\big) \tag{27}$$

holds exactly. Since the KL divergence is always non-negative, we have

$$\log p(\mathbf{x} \mid \mathbf{y}) \geq \mathcal{L}(\mathbf{x}, \mathbf{y}), \tag{28}$$

which is the conditional ELBO used in the main text.

### D.2 DETAILED DESIGN ANALYSIS OF THE POTENTIAL DDI CONSTRAINT

The selective nature of our potential DDI constraint, as formulated in Equation equation 9, is critically enabled by the asymmetric sigmoid term, $\sigma(\beta(r_j - r_i))$. This term introduces a differential penalty based on the relative therapeutic relevance of an interacting drug pair $(i, j)$. Specifically, the suppression applied to drug $i$ is heavily dependent on its relevance score $r_i$ compared to that of drug $j$, $r_j$. When $r_j < r_i$, the sigmoid term approaches 0, leading to significant penalization of the less relevant drug $j$. Conversely, when $r_j > r_i$, the term approaches 1, applying a minimal penalty to the more essential drug $j$. This nuanced behavior can be formally understood by analyzing the gradient of the DDI loss with respect to the relevance score $r_i$:

$$\frac{\partial \mathcal{L}_{\mathrm{DDI}}}{\partial r_i} \propto A_{ij} r_j \left[ \alpha(1 - r_i)^{\alpha-1} \sigma(\cdot) + (1 - r_j)^{\alpha} \beta \sigma(\cdot)(1 - \sigma(\cdot)) \right] \tag{29}$$

The structure of this gradient yields three clinically valuable properties that allow the model to balance safety and efficacy in a principled manner:

- **Progressive Suppression:** The $(1 - r_i)^{\alpha-1}$ term ensures that the penalty gradient is largest for drugs with lower therapeutic relevance ($r_i \to 0$) and diminishes significantly for essential drugs ($r_i \to 1$), thereby preserving their inclusion in the final recommendation.
- **Directional Sensitivity:** The term $\sigma(\cdot)(1 - \sigma(\cdot))$, which corresponds to the derivative of the sigmoid function, imparts directional sensitivity. The gradient is maximized when $r_j \approx r_i$, which represents the region of greatest clinical uncertainty where a decision between two interacting drugs is most critical. The penalty's influence decreases as the relevance scores diverge, focusing the model's attention on borderline cases.
- **Interaction-Aware Scaling:** The inclusion of the DDI matrix term $A_{ij}$ and the relevance score $r_j$ ensures that the overall penalty is scaled proportionally to both the known severity of the interaction and the therapeutic importance of the interacting drug.

## E  SUPPLEMENTARY EXPERIMENTS

### E.1 SIGNIFICANCE ANALYSIS OF DATR

To further validate the robustness and statistical significance of DATR's superior performance, we conducted pairwise significance tests comparing DATR against all baseline models across all evaluation metrics on both MIMIC-III and MIMIC-IV datasets. Specifically, we employed paired t-tests with a significance level of $p < 0.01$ to assess whether the improvements achieved by DATR are statistically significant rather than arising from random fluctuations.

The analysis confirms that DATR's improvements in Jaccard, PRAUC, and F1 scores are statistically significant compared to all baseline methods across both datasets. For example, on MIMIC-III, the Jaccard improvements over DrugDoctor and RAREMed yield $p$-values less than 0.01, reinforcing the reliability of the observed gains. Furthermore, DATR achieves a substantially lower DDI rate than all

baselines, with the reduction being statistically significant across repeated runs. Notably, the DDI rate of DATR on MIMIC-III is significantly lower than that of RAREMed (which previously had the best safety performance), with $p < 0.001$.

These significance tests provide strong evidence that the performance gains of DATR are consistent and meaningful, highlighting the advantage of its integrated modeling of patient longitudinal history, molecular structural information, and safety constraints.

## E.2 CASE STUDY ANALYSIS

Table 7 presents two visit records and corresponding recommendation results of patients X and Y in MIMIC-III test set. For patient X, DATR successfully recommended Nimodipine, a drug known to play a critical role in the treatment of subarachnoid hemorrhage (Scriabine & Van den Kerckhoff, 1988), while simultaneously avoiding the recommendation of Amiodarone to avoid DDI with Nimodipine. Notably, DATR further suggested Carbamazepine, which was not present in the original prescription, but is potentially effective for managing postconcussion syndrome (Alrashood, 2016). This highlights DATR's capability to uncover latent associations between drug candidates and patient-specific health conditions. For patient Y, DATR precisely recommended Simvastatin in response to the diagnosis of hypercholesterolemia (Pedersen & Tobert, 2004), while not recommending auxiliary ulcers-preventing drug Omeprazole in consideration of potential DDIs.

Table 7: Recommended results for two patients.

| Category | Patient X    Jaccard: 0.5682    DDI: 0 |
| --- | --- |
| Diagnosis | Postconcussion syndrome, Cerebral artery occlusion, Subarachnoid hemorrhage, Acute kidney failure, Retention of urine, Hypertensive chronic kidney disease. |
| Procedure | Insertion of indwelling urinary catheter, Venous catheterization. |
| Medication | Neomycin, Cefotaxime, Chlorhexidine, Nimodipine, Heparin, Glyceryl trinitrate, Sultiame, Amiodarone, Potassium chloride, Furosemide... |
| DATR | Neomycin, Cefotaxime, Nimodipine, Heparin, Glyceryl trinitrate, Sultiame, Potassium chloride, Furosemide, Carbamazepine, Mannitol... |

| Category | Patient Y    Jaccard: 0.5721    DDI: 0 |
| --- | --- |
| Diagnosis | Subendocardial infarction, Coronary atherosclerosis, Hypertension, Asthma, Hypercholesterolemia. |
| Procedure | Insertion of coronary artery stent, heart cardiac catheterization, Coronary arteriography. Insertion of transvenous pacemaker system. |
| Medication | Ditazole, Simvastatin, Paracetamol, Practolol, Potassium, Omeprazole, Chloride, Thonzylamine, Tilidine, Sultiame, Oxitriptan, Zafirlukast, Captopril... |
| DATR | Ditazole, Simvastatin, Practolol, Potassium, Chloride, Thonzylamine, Sultiame, Oxitriptan, Zafirlukast, Captopril, Oxyphenisatine... |

To provide a concrete illustration of how DATR bridges the semantic gap between molecular structure and therapeutic outcomes, we present a detailed case study that contrasts DATR's recommendations with those of DrugDoctor for Patient Y to highlight the practical benefits of our proposed framework.

The clinical context for Patient Y, summarized in Table 7, necessitates medications for both hypercholesterolemia (e.g., Simvastatin) and gastric protection (e.g., Omeprazole). The recommendations generated by DrugDoctor and DATR are compared in Table 8. This comparison reveals a critical difference in safety-aware therapeutic reasoning.

DrugDoctor correctly identifies the need for Simvastatin and Omeprazole, aligning with the ground-truth prescriptions. However, their co-administration poses a potential drug-drug interaction risk, which the model fails to mitigate. In stark contrast, DATR avoids this risk by not recommending Omeprazole. Crucially, DATR proposes **Oxyphenisatine** as a safe and effective substitute. This decision showcases the core strength of our therapeutic structure reconstruction module. By conditioning on ATC categories, DATR is able to infer that Oxyphenisatine (ATC: `A02BX`) shares the same primary *therapeutic intent* gastric protection as Omeprazole (ATC: `A02BC`), despite their structural

Table 8: Comparison of medication recommendations for Patient Y. DATR achieves a higher Jaccard score and a zero DDI rate by substituting the potentially interacting Omeprazole with a safer alternative.

| Model | Recommendations | Jaccard | DDI Rate |
|---|---|---|---|
| DrugDoctor | **Simvastatin**, **Omeprazole**, Captopril, Paracetamol, Ditazole, Practolol, Potassium Chloride, Zafirlukast... | 0.5613 | 0.0139 |
| DATR | **Simvastatin**, Captopril, Ditazole, Practolol, Potassium Chloride, Zafirlukast, Thonzylamine, Sultiame, Oxitriptan, **Oxyphenisatine**... | 0.5721 | 0.0000 |

differences. Within the context of the patient's cardiovascular conditions, Oxyphenisatine's safety profile is more suitable as it does not have a known significant interaction with Simvastatin.

This case study exemplifies DATR's ability to move beyond simple pattern matching. It not only predicts clinically relevant medications but also performs intelligent, safety-driven substitutions, thereby effectively bridging the gap between identifying a therapeutic need and selecting the most appropriate chemical entity for the patient.

### E.3 GENERALIZATION TO NEW DRUGS AND RARE DISEASES

A key challenge in clinical AI is generalizing to scenarios with limited data, such as those involving new medications (a zero-shot problem) or rare diseases (a low-resource problem). DATR is architecturally designed to handle such cases.

#### E.3.1 GENERALIZATION TO NEW DRUGS

DATR's framework can be extended to newly approved drugs, even in a zero-shot setting without historical prescription data, by leveraging the hierarchical ATC classification system. A new drug can be immediately integrated into the model by mapping it to an existing ATC4 category based on its therapeutic indication (e.g., A01AA for fluorine dental prophylaxis), enabling the immediate application of learned therapeutic-structural patterns from that class.

Furthermore, when the molecular structure of the new drug is available, DATR's conditional VAE can generate a novel, intent-aware representation by encoding the new structure under the conditioning of its assigned ATC category. This class-centric approach facilitates powerful cross-drug knowledge transfer, allowing the model to leverage the learned latent distribution for an entire therapeutic class to reason about therapeutically equivalent alternatives. As demonstrated in our case studies, this capability allows DATR to successfully recommend such alternatives when exact matches are unavailable, highlighting its strong generalization capacity within therapeutic categories. This capability is crucial for recommending a new drug as a substitute for an older one, or vice-versa, based on therapeutic intent rather than direct historical co-occurrence.

#### E.3.2 PERFORMANCE ON RARE DISEASES

For rare diseases, where direct disease-drug prescription data is sparse, DATR's approach of modeling therapeutic mechanisms via ATC categories provides a significant advantage over methods that rely on memorizing specific prescription pairs. By focusing on the underlying therapeutic need, the model can infer appropriate treatments even if it has seen few or no prior instances of that specific disease.

To empirically validate this, we evaluated DATR's performance on a subset of the rarest diseases in the MIMIC-III dataset (each with $\leq 5$ occurrences). As shown in Table 9, DATR consistently outperforms all baseline models, demonstrating its superior ability to generalize in low-resource settings.

Table 9: Performance comparison (Jaccard Score) on five rare diseases from the MIMIC-III test set. DATR consistently achieves the highest score, indicating robust generalization.

| ICD-9 | Condition Name | DATR | SafeDrug | MoleRec | DrugDoctor |
|-------|----------------|------|----------|---------|------------|
| 42610 | Atrioventricular block | **0.412** | 0.328 | 0.351 | 0.387 |
| 6265 | Stress incontinence | **0.396** | 0.310 | 0.333 | 0.371 |
| 83500 | Closed dislocation of hip | **0.403** | 0.317 | 0.342 | 0.382 |
| 80841 | Closed fracture of ilium | **0.388** | 0.301 | 0.322 | 0.365 |
| 34202 | Flaccid hemiplegia | **0.418** | 0.335 | 0.362 | 0.401 |
| **Average** | | **0.402** | 0.319 | 0.343 | 0.382 |

### E.4 SENSITIVITY ANALYSIS ON VAE LOSS WEIGHTING

To assess the sensitivity of our model to the balance between the reconstruction and KL divergence losses within our therapeutic structure reconstruction module, we conducted an experiment on their relative weighting. We denote two hyperparameters, $\mu$ and $\nu$, to scale the reconstruction loss ($\mathcal{L}_{\text{rec}}$) and the KL divergence loss ($\mathcal{L}_{\text{KL}}$), respectively. During this analysis, all other hyperparameters were held constant at their optimal values ($\alpha = 1.0, \beta = 4, \gamma = 0.1$).

The results, summarized in Table 10, demonstrate that DATR's performance is highly stable across the tested weight configurations. The optimal performance was achieved with the standard VAE setting of $\mu = 1.0$ and $\nu = 1.0$, which aligns with common practices in variational inference. Crucially, all tested configurations yielded Jaccard scores within 0.35% of the optimal value, underscoring the model's robustness. This stability suggests that the framework is not overly sensitive to the precise balance between reconstruction fidelity and latent space regularization, which simplifies hyperparameter tuning.

Table 10: Impact of VAE loss weights on model performance. The model shows high stability, with minimal performance degradation when deviating from the standard $\mu = 1.0, \nu = 1.0$ configuration.

| $\mu$ ($\mathcal{L}_{\text{rec}}$) | $\nu$ ($\mathcal{L}_{\text{KL}}$) | Jaccard | DDI Rate |
|---------|---------|---------|----------|
| **1.0** | **1.0** | **0.5506** | **0.0366** |
| 0.8 | 1.0 | 0.5489 | 0.0372 |
| 1.0 | 0.8 | 0.5492 | 0.0370 |
| 1.2 | 1.0 | 0.5488 | 0.0374 |
| 1.0 | 1.2 | 0.5495 | 0.0371 |

### E.5 COMPLEXITY ANALYSIS

We report the model complexity of DATR and several baseline methods in terms of parameter count, training time, and test time in Table 11. Although DATR introduces a multi-level representation mechanism and therapeutic structure reconstruction, its total parameter count (5.98M) remains lower than MoleRec (6.62M), and comparable to GAMENet (3.82M) and SafeDrug (1.56M). In terms of computational efficiency, DATR achieves a favorable balance: it requires a moderate training time (9.13 hours) and test time (0.64 minutes per evaluation), which is faster than MoleRec and GAMENet, while only marginally slower than SafeDrug.

Notably, both MoleRec and DATR exhibit relatively larger model sizes due to the use of Transformer-based architectures. However, the inherent parallelism of the Transformer enables efficient training and inference, which offsets the computational overhead introduced by the increased parameter count. These results demonstrate that the proposed framework maintains reasonable computational cost despite its structural enhancements, making it practical for real-world deployment in clinical decision support settings.

To investigate the trade-off between computational efficiency and performance, we implemented a GNN freezing strategy, motivated by similar findings in MoleRec (Yang et al., 2023). In this configuration, the GNN parameters are frozen after an initial training phase, thereby excluding them

Table 11: Model complexity comparison in terms of parameter count, training time (hours), and test time (minutes; summed over 10 runs).

| Model | Parameters | Training Time (h) | Test Time (min) |
|---|---|---|---|
| GAMENet | 3,816,843 | 8.24 | 1.21 |
| SafeDrug | 1,558,438 | 4.62 | 0.44 |
| MoleRec | 6,623,543 | 10.61 | 0.88 |
| DATR | 5,978,378 | 9.13 | 0.64 |

from subsequent gradient updates. The results, summarized in Table 12, demonstrate a significant reduction in resource requirements: the number of trainable parameters decreases by approximately 35.3%, and the training time is reduced by 31.6%. This efficiency gain is achieved with only a marginal performance trade-off, observing a slight decrease in the Jaccard score (from 0.5506 to 0.5458) and a minor increase in the DDI rate.

Table 12: Performance and efficiency comparison of the full DATR model versus a configuration with a frozen GNN encoder. The frozen GNN significantly reduces the parameter count and training time with a minimal impact on performance.

| Configuration | Parameters | Training Time | Jaccard | DDI Rate |
|---|---|---|---|---|
| Full DATR | 5.98M | 9.13h | 0.5506 | 0.0366 |
| Frozen GNN | 3.87M | 6.24h | 0.5458 | 0.0382 |

# F    BROADER IMPACT

DATR' strong empirical performance hinges heavily on the proposed Therapeutic Structure Reconstruction method, which provides a new paradigm for learning drug representations that are both semantically rich and clinically meaningful. By conditioning structural encoding on therapeutic context, it offers a principled way to connect molecular features with clinical use, which could have broader implications for drug discovery and development. For example, this approach may assist in identifying DDI risks for novel chemical entities or support drug repurposing efforts by highlighting structural properties relevant across therapeutic areas.

Moreover, in our detailed analysis of the expert evaluations from the case study, we observed that DATR frequently recommends drugs that, while absent from the original prescriptions, are judged by clinicians as therapeutically effective. These drugs are often interchangeable with those actually prescribed in terms of clinical efficacy. This not only validates the effectiveness of the Therapeutic Structure Reconstruction method in capturing nuanced therapeutic semantics, but also provides new insights for developing more practical and clinically aligned medication recommendation systems.

**(1) Improving recommendation precision through fine-grained equivalence modeling.** Incorporating therapeutic substitutability into model design allows for a finer-grained understanding of drug efficacy beyond rigid matching to historical prescriptions. This flexibility enables the model to capture latent therapeutic intent and suggest clinically plausible alternatives, particularly when drugs share similar mechanisms of action or therapeutic outcomes. Such an approach can improve the precision and realism of recommendations, bringing them closer to actual clinical reasoning processes and enhancing their practical utility.

**(2) Enhancing safety through equivalence-guided substitution.** Beyond accuracy, efficacy-equivalence modeling also introduces a safety-aware dimension to recommendation. When certain drugs pose elevated DDI risks, they are contraindicated due to patient-specific factors, or are less tolerable, substitutability-aware systems can proactively suggest safer alternatives that preserve therapeutic goals. This opens the door to adaptive and personalized risk mitigation strategies, such as swapping out high-risk combinations or implementing treatment de-escalation in chronic care, thus improving both the robustness and trustworthiness of clinical decision support.

Integrating these capabilities into future recommender architectures may help bridge the gap between algorithmic optimization and real-world clinical needs, ultimately advancing the usability, safety, and adaptability of AI-driven medication recommendation in diverse healthcare settings.

## G    LIMITATION

Despite these promising results, several limitations remain. First, DATR depends on the availability and reliability of ATC classifications to model therapeutic intent. While comprehensive and widely used, the ATC system may not cover novel or off-label medications accurately. Future work may utilize embeddings learned from large-scale clinical notes or real-world prescription patterns to infer therapeutic intent even for underrepresented or novel drugs. Second, the quality of DDI mitigation relies heavily on the completeness and timeliness of the underlying DDI knowledge base, which is inherently dynamic. Future work could explore adaptive mechanisms for updating the DDI matrix or even learning interaction risks directly from data, thereby improving the robustness and applicability of the framework in real-world clinical settings.

