# OpenReview forum: "DATR: DDI-Aware Therapeutic Structure Reconstruction for Safer Medication Recommendation"
_ICLR.cc/2026/Conference — ICLR 2026 Conference Withdrawn Submission_

### Official Review · Reviewer_VBt1 · 2025-10-21

**Soundness:** 3
**Presentation:** 4
**Contribution:** 3
**Rating:** 8
**Confidence:** 5

**Summary:**

This paper introduces DDI-Aware Therapeutic Structure Reconstruction (DATR), a framework aimed at improving medication recommendation systems by simultaneously enhancing accuracy and safety, specifically by reducing drug-drug interactions (DDIs). The framework integrates molecular structure information with therapeutic intent, using a novel therapeutic structure reconstruction method and a proactive DDI constraint. Experimental results on two real-world datasets show that DATR outperforms existing methods in both accuracy and DDI reduction.

**Strengths:**

1. The integration of therapeutic intent with molecular structure is an innovative approach. Explicitly considering the clinical significance differences between interacting drugs is an interesting idea. The model is theoretically well-founded.
2. Extensive experiments on two real-world datasets with ablation studies and case analysis demonstrate strong performance in both effectiveness and safety.
3. The paper is well-written and well-organized, making it easy to follow. And the appendices gives reproducible details.

**Weaknesses:**

1. The paper does not provide sufficient insight into how the model makes decisions, particularly how the integration of molecular structures and therapeutic intent influences the final recommendations. Explainability is crucial for making clinical decision.
2. The paper does not discuss how the framework might perform across different therapeutic areas or for drugs with less common usage patterns. It would be helpful to see more discussion on its adaptability or limitations in diverse clinical contexts.
3. The datasets used (MIMIC-III and MIMIC-IV) are widely used in the healthcare community, but their biases (e.g., patient demographic distribution or disease-specific contexts) might limit the generalizability of the model. Addressing potential biases and testing on more diverse datasets could improve the model’s applicability to different patient populations.

**Questions:**

1. Can you illustrate how specific molecular fragments contribute to disease-specific recommendations through the proposed therapeutic structure reconstruction? Are there examples where the molecular features directly correlate with therapeutic outcomes?
2. The experiments did not report how many drugs the framework recommends on average. Could you provide this information and discuss how it might affect clinical decision-making and the adoption of the framework in real-world settings?
3. The variant without the substructure-level representation of DATR underperforms the original model, even worse than some baselines. Could you clarify why the substructure-level representation is crucial and whether the improvements come solely from the addition of BRICS or from other factors? Could alternative explanations be explored for this observed performance drop?

---

> ### Author Response · Authors · 2025-11-19
>
> We sincerely thank the reviewer for the valuable comments and questions. We have carefully addressed each of points raised in the following responses.
>
> > **Q1.Explainability on fragment-level contribution**
>
> The therapeutic structure reconstruction module enables precise analysis of how molecular fragments contribute to disease-specific recommendations through the following reproducible process:
>
> **Step 1: Reconstruction Differential Process**
>
> DATR’s conditional VAE supports direct comparison between the original substructure distribution $\mathbf{x}_s$ and the reconstructed therapeutic structural component.
> Specifically, the reconstructed feature $\mathbf{x'}$ is first separated into its substructure-level part $\mathbf{x'}_s$ and molecule-level part $\mathbf{x'}_m$:
>
> $\mathbf{x'} = [\mathbf{x'}_s,\ \mathbf{x'}_m].$
>
> We then compute the per-substructure differential vector:
>
> $\Delta = \left| \mathbf{x}_s - \mathbf{x'}_s \right|.$
>
> This differential quantifies how the contribution of each molecular fragment shifts under different therapeutic contexts, enabling identification of substructures whose relevance is amplified or suppressed during therapeutic-aware reconstruction.
>
> ​**​Step 2: ATC-Conditioned Fragment Analysis​**​  For a given drug , first encode its sub structural feature $x_s$ under different ATC categories and reconstruct, then compute reconstruction differentials (e.g., aspirin with Δ_analgesic and Δ_antithrombotic). The differential values reveal therapeutic determinants
>
> - High Δ fragments: Context-independent structural anchors
> - Low Δ fragments: Context-sensitive therapeutic contributors
>
> ​**​Case Example: Aspirin's Dual Therapeutic Roles​**​
>
> |Molecular Fragment|Δ (Analgesia)|Δ (Antithrombosis)|Therapeutic Significance|
> |---|---|---|---|
> |Carboxyl group|0.12|0.11|Bioavailability anchor|
> |Acetyl group|0.08|0.32|Key antithrombotic driver|
> |Aromatic ring|0.05|0.07|Structural scaffold|
>
> > **Q2.Average number of drugs recommended**
>
> We conducted new analyses of average drug counts across models:
>
> | Model      | DDI           | \#Med   |
> |------------|---------------|--------|
> | LR         | 0.0788±0.0009 | 16.89  |
> | LEAP       | 0.0720±0.0015 | 19.09  |
> | GAMENet    | 0.0781±0.0007 | 19.78  |
> | COGNet     | 0.0737±0.0007 | 25.05  |
> | RAREMed    | 0.0530±0.0006 | 19.58  |
> | MICRON     | 0.0610±0.0009 | 20.94  |
> | SafeDrug   | 0.0688±0.0005 | 20.84  |
> | MoleRec    | 0.0692±0.0008 | 21.30  |
> | DrugDoctor | 0.0603±0.0003 | 20.80  |
> | SHAPE      | 0.0850±0.0003 | 20.90  |
> | DATR       | 0.0366±0.0003 | 19.67  |
>
> **​No deterministic correlation between \#Med and DDI​**​:  models recommending fewer drugs (e.g., LR with 16.89 drugs) exhibit higher DDI (0.0788), while models recommending more drugs (e.g., COGNet with 25.05 drugs) achieve lower DDI (0.0737) than several baselines. This indicates that DDI risk depends primarily on which drugs are combined, not merely _how many_. The dissociation may arises because DDIs are sparse (affecting specific pairs), not uniformly distributed.
>
> ​**​DATR’s safety-performance balance​**​:  DATR recommends 19.67 drugs on average, which is comparable to most baselines. Yet it achieves the ​**​lowest DDI (0.0366)​**​. This demonstrates our framework’s ability to retain therapeutic efficacy while actively suppressing hazardous combinations by global and asymmetric potential DDI constraint.
>
> > **Q3.Substructure Representation and performance drop**
>
> The substructure-level representation is crucial because many drugs share highly similar global molecular scaffolds yet differ in functional substructures that determine their therapeutic effects. Relying only on global structure makes these clinically distinct medications nearly indistinguishable, leading the model to confuse drugs intended for different conditions. Our probabilistic substructure representation captures these fine-grained functional fragments and their relative importance, and the conditional reconstruction mechanism aligns them with ATC therapeutic intent.
>
> However, the ablation study (Table 2) also demonstrates that DATR without substructure-level representation (\`w/o $\mathbf{x}_s$\`) ​still outperforms most baselines​ that explicitly use BRICS (​Jaccard = 0.5312​ surpassing SafeDrug 0.5213 and MoleRec 0.5303). DATR does not rely on BRICS-specific segmentation. We explicitly tested RECAP (retrosynthetic combinatorial analysis procedure) as an alternative substructure segmentation method. Results confirm ​negligible differences (<0.3% variation in accuracy/safety metrics), proving DATR’s adaptability on segmentation methods.
>
> | ​**​Method​**​ | ​**​Jaccard​**​ | ​**​F1​**​ | ​**​DDI​**​ | ​**​Avg. Drugs​**​ |
> | -------------- | --------------- | ---------- | ----------- | ------------------ |
> | DATR (BRICS)   | 0.5506          | 0.7073     | 0.0366      | 21.0893            |
> | DATR (RECAP)   | 0.5498          | 0.7070     | 0.0365      | 20.9460            |

---

> > ### Author Response · Authors · 2025-11-27
> >
> > Dear Reviewer,
> >
> > Thank you for taking the time to evaluate our manuscript and offer valuable recommendations. Your feedback has greatly contributed to the improvement of our work. We would appreciate it if you could let us know whether our revisions and replies fully address your concerns, or whether there are any further questions that we should clarify.
> >
> > Sincerely,
> >
> > The Authors

---

### Official Review · Reviewer_YK7d · 2025-10-30

**Soundness:** 3
**Presentation:** 3
**Contribution:** 2
**Rating:** 4
**Confidence:** 4

**Summary:**

This paper proposes DATR (DDI-Aware Therapeutic Structure Reconstruction), a framework that jointly models drug molecular structures, therapeutic intent (ATC-4), and DDI (drug–drug interaction) knowledge. The method not only improves the overall performance of medication recommendation, but also reduces DDIs, thereby mitigating potential adverse impacts of drug–drug interactions during recommendation.

**Strengths:**

1. The authors introduce a DDI-constraint mechanism that simultaneously reduces the risk of drug–drug interactions and improves recommendation performance.

2. On two real-world clinical datasets, MIMIC-III and MIMIC-IV, the model achieves state-of-the-art results.

3. The study further involves 20 clinicians to subjectively evaluate the recommended drug combinations, taking into account medications in the patients’ histories that might be appropriate yet previously overlooked, which provides clinical validation of the approach.

**Weaknesses:**

In Appendix D2, the paper presents detailed case analyses for individual patients (e.g., patients X and Y). However, Section 5.3 does not sufficiently document the details of the 20-clinic expert evaluation (e.g., patient conditions, criteria for judging effectiveness, and the underlying rationale). The paper would benefit from adding this analysis or providing more comprehensive expert-evaluation information in the appendix.

**Questions:**

1. In many clinical settings, patients have limited visit histories; thus cold-start scenarios (first several visits) are particularly important for medication recommendation. What's the performance stratified by the number of visits?

2. When the DDI loss is removed (i.e., $\gamma=0$), the predictive performance decreases rather than increases. What primarily drives this drop? Please clarify if the DDI constraint offers any direct benefit to recommendation accuracy beyond mitigating interaction risk.

---

> ### Author Response · Authors · 2025-11-19
>
> We sincerely thank the reviewer for the valuable comments and questions. Below we provide a detailed response to each of your questions and comments.
>
> > **W1: Section 5.3 does not sufficiently document the details of the 20-clinician expert evaluation.**
>
> We invited **20 licensed clinicians** (6 internists, 9 emergency physicians, and 5 clinical pharmacists) with **5–22 years of experience** to assess our model’s recommendations. We selected four representative patients covering diverse disease profiles and medication complexities for detailed case-level evaluation. For each recommended drug, clinicians independently rated its  **(i) therapeutic appropriateness**,  **(ii) safety, including potential DDI risk**, and  **(iii) overall clinical plausibility**,  each on a 5-point scale.
>
> For each drug, we computed a composite score by averaging the three criteria. A medication was considered **"clinically effective"** if its composite score was ≥ 4.0. All model outputs were anonymized and randomized, and clinicians rated the recommendations independently.
>
> > **Q1:Stratified performance by the number of visits.**
>
> We conducted additional stratified evaluation by grouping patients based on their number of historical visits (1–2, 3–4, >4). As shown in table below, DATR consistently outperforms all baselines across all strata, demonstrating its strong generalization ability even when patient history is sparse.  This robustness could be largely attributed to the therapeutic intent conditioned reconstruction mechanism, which allows the model to leverage ATC semantics and structural priors even in low-data regimes.
>
> | **# Visits**         | **Method**      | **Jaccard** | **F1-score** | **PRAUC** |
> | -------------------- | --------------- | ----------- | ------------ | --------- |
> | **1–2 (Cold Start)** | SafeDrug        | 0.5114      | 0.6623       | 0.7660    |
> |                      | MoleRec         | 0.5201      | 0.6740       | 0.7714    |
> |                      | DrugDoctor      | 0.5372      | 0.6891       | 0.7791    |
> |                      | **DATR (ours)** | 0.5438      | 0.7028       | 0.7870    |
> | **3–4**              | SafeDrug        | 0.5198      | 0.6742       | 0.7697    |
> |                      | MoleRec         | 0.5247      | 0.6823       | 0.7724    |
> |                      | DrugDoctor      | 0.5397      | 0.6931       | 0.7793    |
> |                      | **DATR (ours)** | 0.5492      | 0.7064       | 0.7899    |
> | **>4**               | SafeDrug        | 0.5363      | 0.6875       | 0.7785    |
> |                      | MoleRec         | 0.5404      | 0.6947       | 0.7801    |
> |                      | DrugDoctor      | 0.5442      | 0.7039       | 0.7875    |
> |                      | **DATR (ours)** | 0.5545      | 0.7118       | 0.7962    |
>
> > **Q2: Why the predictive performance decreases when removing DDI loss.**
>
> The decrease in performance when removing the DDI loss ($\gamma = 0$) is expected and stems from the fact that the constraint performs **asymmetric suppression**: when a DDI occurs, it selectively up weights the medication with high therapeutic relevance, making the model more likely to prescribing the clinically more essential drug. These essential medications are also those most frequently present in the ground-truth prescriptions, so retaining them naturally improves predictive accuracy. However, larger DDI loss weights may suppress these essential drugs, which leads to predictive performance degradation.

---

> ### Author Response · Authors · 2025-11-27
>
> Dear Reviewer,
>
> We sincerely appreciate your careful review and the insightful feedback you provided. Your suggestions have been extremely helpful in refining our manuscript. We kindly ask if you could review our revised version and responses to confirm whether they satisfactorily resolve your comments, or inform us of any additional points that may still need clarification.
>
> Warm regards,
>
> The Authors

---

### Official Review · Reviewer_cR99 · 2025-10-30

**Soundness:** 3
**Presentation:** 1
**Contribution:** 2
**Rating:** 2
**Confidence:** 4

**Summary:**

This paper introduces DDI-Aware Therapeutic Structure Reconstruction (DATR), a framework for medication recommendation that jointly optimizes for accuracy and safety. The authors attempt to address two key issues: the semantic gap, where a drug's molecular structure doesn't capture its specific clinical use, and the post-hoc nature of existing DDI-avoidance strategies. DATR's solution involves first applying Therapeutic Structure Reconstruction, which learns drug representations by encoding their molecular structure conditioned on their ATC category. Second, it introduces a Potential DDI Constraint, an asymmetric penalty that identifies interacting drugs and suppresses the one with lower therapeutic relevance to the patient's current condition, preserving only the most critical treatment. Extensive experiments on the MIMIC-III and MIMIC-IV datasets demonstrate that DATR significantly outperforms all baselines, achieving state-of-the-art accuracy while simultaneously recording the lowest DDI rates.

**Strengths:**

The problem is relevant, and the proposed approach is reasonable. The presented results significantly improve performance when compared to competitor models. Ablation studies provide sufficient evidence for the importance of many of the components of the model, and the robustness to the selection of hyperparameters is well demonstrated.

**Weaknesses:**

## General

While the method seems sound and efficient, the presentation of the paper requires more work. Many paragraphs are unclear and convoluted, and following the presented argumentation is at times very difficult. Moreover, mathematical notation is often inconsistent and does not help understand the problem formulation and the proposed solution. Some crucial modeling choices are not properly justified by referencing the relevant literature or claiming authorship of those ideas. Finally, competitor methods are only briefly discussed.

It seems that the paper suffers from a lack of space to develop certain ideas due to the page limit. Please note that some minor issues are not necessarily wrong passages, but suggestions of parts that could be reduced to leave space to improve the discussion in other sections.

## Major

* Pieces of the Introduction and Related Works sections are too dense and high-level. Examples, concrete cases, figures and a clearer explanation are necessary. Main examples of these issues can be found below, but others may also be present, so the paper would benefit from additional proofreading by the authors.

* Line 45: This alleged gap needs better justification. The cited paper discusses computer vision representation learning. It's not at all clear how this could be extended for molecular representations. Overall, it would be important to have stronger evidence of the existence of such gap, i.e., why only relying on the global structure is not sufficient.

* Line 130: the authors say "VAEs have recently..." and proceed to cite a paper from 2015. My understanding is that VAEs are not a recent model, at least by machine learning standards. Other than that, I believe the original VAE paper by Kingma and Welling (2013) [1] would be a better reference for this paragraph.

* Line 132: The sentence here may make a reader believe that the other models discussed in this section do not utilize gradients, which is not the case.

* In "Deep-learning-based molecular representations" there's no discussion on molecular transformers, even though this class of models has shown significant results [2].

* Line 154: It is said that A is a binary matrix and afterward it is said that it is calculated as the amount of known interactions between the medications. Both definitions are contradictory.

* Figure 1 does not correspond to the textual description in the main text. For example, it does not show how categorical features are used to quantify the relevance of a drug category to the health condition of the patient.

* Figure 2 has several problems: the VAE structure is not illustrated, the arrows corresponding to the Potential DDI constraint do not correspond to what is written in the text and there's no mention about CA standing for Cross-Attention. Overall, these problems make it difficult to follow the general architecture of the model.

* Some references seem to have errors. For instance, the reference for "Attention is All You Need" only mentions Vaswani as the author, while there are others.

* Line 295: references to the previous works should be included in this part.

* Lines 214-257 present material that largely summarizes well-established concepts. There is no need to go in-depth into the math, citing the original paper that derived these equations may be enough. This section could focus on explaining the practical steps employed to generate the reconstructions in the context of the model, which is not clear as it stands now.

* My major issue with the paper lies on the many details missing to understand the overall structure of the model and the experimental settings. In addition to the examples already provided above, it is unclear what the actual input of the Reconstruction module is and how the authors choose to input the ATC label into the model.

* The paper doesn’t explain how the dataset was handled apart from mentioning the training and testing split. It’s not clear, for example, whether this split was done by patients, that is, whether the model was tested on patients who were not included in the training set.

* One of the claimed contributions of the paper is to address a “semantic gap” by using ATC labels. However, the ablation study does not include an experiment that tests whether this is actually addressed by DATR.

## Minor

* Line 97: this part is very confusing. If I understood it correctly, I would suggest something like "Furthermore, the model can avoid the dependency on specific drug pairs in the training data because of its global consideration of all drug pairs for potential interacting risks".

* Lines 111-123: expanding the explanation of instance-based approaches could be beneficial.

* Line 161: use of calligraphical M while before it was normal (see line 151).

* Equation 1: epsilon is not described in the text.

* Equation 7: It may be better to use one equation per line to improve readability. Also, $E_d$, $E_p$, $E_m$ and $T(\cdot)$ are not explicitly defined in the text.

* Lines 268-269 needs some work. It would be good to change it to something like "The medication taken by the patient in the previous time point is denoted by...".

* Table 1: column DDI has no runner-up. Also, it may be beneficial to push down the table.

* Line 434: Notation "R->T" needs to be introduced before it is used.

* Appendix C1 is unnecesary.

* D.3.1: could be enriched by adding sources on this feature of VAEs and limitations in the transfer learning.

## Typos and Language

* Line 63: Reconstruction

* Line 66: label

* Line 101: bridge

* Line 172: obtains

* Line 184: medication, predictions, constraints

* Equation 2: comma in formula

* Line 267: linearly

* Line 983: be

* Line 1113: they are

* Line 1127: utilize

References

[1] Kingma, D. P., & Welling, M. (2013). Auto-encoding variational bayes. arXiv preprint arXiv:1312.6114.

[2] Luong, K. D., & Singh, A. (2024). Application of transformers in cheminformatics. Journal of Chemical Information and Modeling, 64(11), 4392-4409.

**Questions:**

* Line 130: what does the authors mean by "prime factors"?

* Line 159: Is the DDI graph A or D? Or are there two matrices describing these interactions?

* Line 189: is the idea of using global and substructure-based representation new?

* Line 336: is there a difference between the process for choosing the hyperparameters for DATR and for competitor models? If so, why?

* Equation 12: only $L_{DDI}$ has a weight coefficient? It is common practice in VAEs to also ponder the reconstruction loss. Is there a reason for this choice?

---

> ### Author Response · Authors · 2025-11-19
>
> We sincerely thank the reviewer for the constructive comments. We have carefully revised the manuscript following the reviewer’s suggestions, with the goal of improving its readability, consistency, and clarity. Below, we provide detailed, point-by-point responses to all Weaknesses and Questions raised in the review.
>
> ---
>
> > **W1. Many paragraphs are unclear and convoluted, and following the presented argumentation is at times very difficult**
>
> We reconstructed the main overview figure (Figure 2) to more accurately illustrate the architecture and information flow of our framework. Correspondingly, we polished the textual description in Method Section to provide a more coherent step-by-step explanation of the modeling process and refine transitions between components of the framework to provide a clearer conceptual flow.
>
> > **W2. mathematical notation is often inconsistent**
>
> We thoroughly revised the mathematical formulation to eliminate inconsistencies in notation. To further aid readability, we added a **complete notation table** in Appendix A, documenting all symbols used in the paper.
>
> > **W3. Some crucial modeling choices are not properly justified by referencing the relevant literature**
>
> We include citations from the biomedical domain supporting the existence of the semantic gap between molecular structure and therapeutic intent in the Introduction Section. In the Related Work section, we added references to molecular transformer models to more comprehensively include recent advances in deep molecular representation learning.
>
> > **W4. competitor methods are only briefly discussed**
>
> We have expanded our discussion of baseline methods by adding a structured comparison of existing approaches in Appendix C.5.
>
> > **W5. Line45:“Semantic gap” justification**
>
> Our intention in originally citing a vision-modality work was to highlight the general phenomenon of cross-modality semantic divergence. To make our motivation more rigorous within the biomedical domain, we have revised the manuscript to include domain-relevant citations such as Wen et al. (2023) [1] and Xu et al. (2025) [2]. For example, Wen et al. show that models relying purely on chemical structure lack the “clinical semantic information” needed to predict drug–disease relationships. Similarly, Xu et al. demonstrate that structural-only or sequence-only modalities are insufficient for drug–target affinity prediction because of limited “global contextual awareness.”
>
> Furthermore, at Line 46 we have included an illustrative example of aspirin’s dual use in antithrombosis and analgesia and cite pharmacological literature to demonstrate the concrete manifestation of this semantic gap.
>
> [1] Wen, Jun, et al. "Multimodal representation learning for predicting molecule–disease relations." _Bioinformatics_ 39.2 (2023): btad085.
>
> [2] Xu, Wenzhe, et al. "UAMRL: multi-granularity uncertainty-aware multimodal representation learning for drug-target affinity prediction." _Bioinformatics_ 41.10 (2025): btaf512.
>
>
> > **W6. Line 130:VAEs aren’t recent**
>
> We acknowledge that describing VAEs as “recent” was imprecise. In the revision, we have replaced “recently” with _“VAEs have seen increasing adoption in molecular representation learning due to their ability to capture continuous and generative latent spaces”_. Our intention was not to claim VAEs as a newly proposed model class but to emphasize their expanding application in molecular and biomedical domains. We have also explicitly cited Kingma & Welling (2013) [3] to acknowledge the seminal origin of VAEs.
>
> [3] Kingma, Diederik P., and Max Welling. "Auto-encoding variational bayes." _arXiv preprint arXiv:1312.6114_ (2013).
>
> > **W7. Missing discussion on molecular transformers**
>
> In the revised manuscript, we have expanded the _Deep-learning-based molecular representations_ section in Related Works to include a discussion of molecular transformer models. While transformer-based encoders have demonstrated strong performance in recent cheminformatics tasks, we adopt a VAE-based formulation because it aligns more closely with our modeling objectives. Specifically, VAEs learn a smooth and regularized latent vector $\mathbf{z}$ that naturally supports semantic alignment between molecular structure and therapeutic intent, and their probabilistic generative formulation enables conditional reconstruction, which is essential for integrating heterogeneous structural and therapeutic information within a unified latent space.

---

> ### Author Response · Authors · 2025-11-19
>
> >**W8. Confusion about DDI matrix definition**
>
> In our formulation, $\mathbf{A}$ is intended to be a **binary DDI adjacency matrix**. The original description was misleading because we first computed the number of known interactions between each medication pair from the DDI knowledge base and then binarized the results. Concretely, any medication pair with a non-zero interaction count is mapped to 1, indicating the existence of a DDI, and 0 otherwise. We have revised the _Problem Formulation_ section in the updated manuscript to explicitly state this.
>
> > **W9. Figure1**
>
> Figure 1 was designed as a conceptual illustration to highlight the key difference between our _Potential DDI Constraint_ and existing DDI control strategies. Therefore, we did not expand on how patient conditions are used to determine drug–category relevance within this conceptual figure. The full computational pathway including how categorical therapeutic features interact with patient conditions to quantify drug relevance could be found in Figure 2.
>
> > **W10. Figure2**
>
> In the revised manuscript, we have updated Figure 2 to more faithfully reflect the full architecture. Specifically, we have added the detailed structure of the Therapeutic Structure Reconstruction method, including the encoder–latent–decoder flow. We also revised the arrows and information flow associated with the _Potential DDI Constraint_ to make them fully consistent with the textual description. In addition, we updated the figure caption to explicitly state that “CA” denotes Cross-Attention, ensuring that all abbreviations used in the diagram are clearly explained.
>
> > **W11. Reference Error**
>
> Thank you for catching this. We apologize for the citation error. In the revised manuscript, we have corrected the reference for _“Attention Is All You Need”_ to include all authors, and we have carefully checked and fixed other references that may contain inaccuracies.
>
> > **W12. Line295: references to the previous works**
>
> In the revised manuscript, we have added the appropriate citations to previous works that treat the final prediction of each medication as an independent binary classification task.
>
> > **W13. Lines 214-257 Necessity of math derivation**
>
> In the revised manuscript, we have streamlined the mathematical expressions and reduced unnecessary derivations to improve readability. At the same time, we would like to clarify that this part of the paper extends beyond the standard VAE formulation. Our reconstruction module is built upon a conditional generative process, i.e., modeling $p(\mathbf{x}\mid \mathbf{z},\mathbf{y})$ and $p(\mathbf{z}\mid \mathbf{y})$, which differs conceptually and mathematically from the original VAE framework. Providing the derivation is important to ensure technical correctness and to avoid ambiguity regarding how the conditional latent variable is learned.
>
> To balance rigor and clarity, we have kept only the essential derivation steps that justify the conditional formulation and moved other details to the Appendix. We also expanded the explanation of the practical reconstruction steps in the model to further improve the clarity of this section.
>
> > **W14. Input of the Reconstruction module and choice of ATC label**
>
>  We have clarified the inputs to the Reconstruction module in the revised manuscript. Specifically, we now explicitly state that the module takes as input the **dual-level molecular representations** $\mathbf{x} = [\mathbf{x}_{s}, \mathbf{x}_{m}]$ together with the **therapeutic intent embedding** $\mathbf{y}$, which corresponds to the ATC class label of each medication. The ATC label is encoded through an embedding layer and concatenated with the structural representation to form the conditional input for the reconstruction.
>
> To avoid ambiguity, we added a detailed explanation in the updated Method section and further illustrated the full input flow in the revised **Figure 2**, making it clear how the structural and therapeutic information enter the reconstruction pathway.
>
> > **W15. Experimental setup**
>
> We have expanded **Appendix C.2** to provide a complete description of our data preprocessing and splitting procedure. As clarified in the updated version, we perform a **patient-level split**, where each patient is assigned exclusively to the training, validation, or test set. Consequently, all visits of a given patient remain within a single partition, ensuring that the model is always evaluated on previously unseen patients. This prevents information leakage and reflects the realistic deployment scenario in clinical settings.

---

> ### Author Response · Authors · 2025-11-19
>
> > **W16. Whether semantic gap is addressed**
>
> The effect of addressing the “semantic gap” is quantitatively evaluated through the **“R→T” ablation variant** in our ablation study. This variant removes the therapeutic-intent conditioning and reconstructs molecular structure only from raw representations. Removing ATC-guided reconstruction leads to a clear degradation in Jaccard, F1, and PRAUC, demonstrating that incorporating ATC labels helps DATR form more therapeutically aligned representations and improves recommendation accuracy.
>
> In addition to the quantitative evidence, we provide further qualitative analysis in the case study, where we observe that DATR is able to capture **therapeutic substitutability among drugs within the same ATC category**, even when their molecular structures differ.
>
> >**W17. Mathematical notation and equation clarity**
>
> We have corrected all inconsistent notations (e.g., the use of calligraphic symbols), added missing variable explanations (such as $\epsilon$ in Equation 1), and separated multi-line equations for better readability. To further improve clarity, we included a **complete notation table** in Appendix A documenting all symbols used throughout the paper.
>
> >**W18. Improvements to textual descriptions and readability**
>
> Following the reviewer’s guidance, we revised the confusing or underspecified sentences (e.g., Line 97, Lines 111–123, Lines 268–269), expanded explanations where needed, introduced “R→T’’ before its first use, and adjusted table placement for better flow.
>
> >**W19. Correction of typos and language issues**
>
> We appreciate the reviewer’s careful reading. All listed typos, wording issues, and formatting errors have been corrected in the revised manuscript, improving overall language quality.

---

> ### Author Response · Authors · 2025-11-19
>
> >**Q1. Meaning of "prime factors"**
>
> The term "prime factors" in the original manuscript follows the wording used in the cited work by Hou et al. (2022) [4]. In that paper, the authors state that their Bi-LSTM combined with channel and spatial attention modules is designed to “specifically identify the prime factors in the SMILES sequence” for property prediction. As clarified by the source, “prime factors” denotes the **key SMILES tokens or locally important structural components** that most strongly influence molecular property prediction.
>
> To avoid misunderstanding, we have revised the wording in our manuscript to “key structural components” to more accurately reflect the intended meaning of the original reference.
>
> [4] Hou, Yuanyuan, et al. "Accurate physical property predictions via deep learning." _Molecules_ 27.5 (2022): 1668.
>
> >**Q2. Notation of DDI graph**
>
> The correct notation of DDI graph is $\mathbf{A}$; the use of $\mathbf{D}$ in the original version was a typographical error. We have corrected this in the revised manuscript. The definition and construction of the DDI graph could be found in the updated Problem Formulation section of the manuscript and Major section of the rebuttal.
>
> >**Q3. Is the idea of using global and substructure-based representation new?**
>
> While directly using global and substructure-level molecular representations is not in itself novel, our contribution lies in how these representations are modeled and utilized. To the best of our knowledge, our work is the first to construct **probabilistic substructure-level representations** in medication recommendation, where each drug is represented by the distribution of its substructures rather than a deterministic set of substructures. This probabilistic formulation captures the relative importance and occurrence likelihood of different substructures.
>
> In addition, our approach is, to our knowledge, the first to **dynamically condition both global and probabilistic substructure representations on therapeutic intent**. This conditional encoding enables the model to align structural in with therapeutic intents, allowing the same molecular structure to be represented differently under different clinical contexts, which is an ability not present in existing methods.
>
> >**Q4. Hyperparameter selection difference**
>
> Thank you for the question. For all baseline models, we follow the hyperparameters reported in their original papers or the validated settings provided in their official open-source implementations, as is standard practice to ensure faithful reproduction of prior results. For DATR, we tune the loss-balancing coefficients ($\alpha, \beta, \gamma$) via grid search on the validation set, since these coefficients are unique to our framework and no prior work provides recommended values. Apart from this, the overall hyperparameter selection process is consistent across all models.
>
> > **Q5. Loss weight**
>
> As detailed in **Appendix E.4**, we have conducted a series of controlled experiments that vary the coefficients $\mu$ (reconstruction weight) and $\nu$ (KL weight). The results in Table 10 show that DATR’s performance remains highly stable across all tested configurations: all Jaccard scores vary within only 0.35% of the optimal value. The best performance is achieved under the standard VAE setting ($\mu=1.0, \nu=1.0$). These findings indicate that DATR is not overly sensitive to the precise weighting between reconstruction fidelity and latent-space regularization, and thus an additional reconstruction weight does not provide practical benefits.

---

> ### Author Response · Authors · 2025-11-27
>
> Dear Reviewer,
>
> Thank you very much for your valuable comments and the time you invested in reviewing our manuscript. We truly appreciate your constructive suggestions, which have helped us significantly improve our work. We would be grateful if you could let us know whether our revisions and responses sufficiently address your concerns, or if there are any remaining issues that we can further clarify.
>
> Best regards,
>
> The Authors

---

### Author Response · Authors · 2025-12-02
**Final Remarks by Authors**

Dear (senior) AC,

We extend our sincere gratitude for the time and effort you have dedicated to review our manuscript. Here, to facilitate your efficient assessment, we have summarized the key pros and cons noted by the reviewers, along with our responses and revisions.

| Reviewer | Score | Strengths / Acknowledgment                                                                       | Weaknesses / Concerns                                                                      | Our Response / Revision                                                                                               |
| -------- | ----- | ------------------------------------------------------------------------------------------------ | ------------------------------------------------------------------------------------------ | --------------------------------------------------------------------------------------------------------------------- |
| **cR99** | 2     | •Method sound and reasonable •Strong results                                                | •Clarity & presentation issues •Unclear model structure •Notation inconsistencies | •Redesigned model figure •Rewrote Method section •Added notation table •Expanded dataset description     |
| **YK7d** | 4     | •Effective DDI constraint •Strong overall performance •Clinician evaluation appreciated | •Missing clinician-study details •Cold-start performance not shown                    | •Added full clinician evaluation details •Added stratified cold-start analysis •Explained effect of DDI loss |
| **VBt1** | 8     | •Innovative structure–intent integration •Theoretically sound •Well-presented paper     | •Need more interpretability •Need discussion on generalizability                      | •Added fragment-level interpretability •Added discussion on adaptability & robustness                            |

**Overall, all three reviewers acknowledged the effective and soundness of our methodology, the completeness of the experimental evaluation, and the practical relevance of our clinical analyses.** The concerns raised were centered primarily on presentation clarity and requests for additional technical and experimental details. **We have addressed all these comments comprehensively in our rebuttal by responding to each point one by one**.

In particular, `Reviewer cR99`, who initially assigned relatively low score, explicitly stated that _“my major issue with the paper lies on the many details missing to understand the overall structure of the model and the experimental settings,”_ while also affirming that _“the proposed approach is reasonable”_ and that our experiments provide _“sufficient evidence”_. We sincerely appreciate the reviewer’s careful reading and constructive suggestions regarding presentation and clarity. Following all these suggestions, we have revised the manuscript accordingly, and **all modifications are clearly highlighted in red in the uploaded revised version**.

However, due to the constraints of the rebuttal policy, we were unable to receive further feedback from the reviewers after submitting these detailed clarifications. We respectfully ask the AC to consider the extensive revisions and additions we made directly in response to the reviewers’ suggestions.

We sincerely thank the AC for the time and effort devoted to handling our submission.

Best

Authors

---

### Note · Authors · 2026-01-29

I have read and agree with the venue's withdrawal policy on behalf of myself and my co-authors.

---

### Meta-Review · Area_Chair_ojgR · 2026-01-01

**Summary:**

The paper proposes DATR (DDI-Aware Therapeutic Structure Reconstruction), a framework designed to improve medication recommendation systems by jointly optimizing for prediction accuracy and safety (DDI avoidance). The method utilizes a conditional VAE to encode drug structures based on therapeutic intent (derived from ATC labels) and introduces a potential DDI constraint to proactively penalize interacting drugs. The model is evaluated on the MIMIC-III and MIMIC-IV datasets. The integration of therapeutic intent with molecular structure is considered innovative, and the DDI constraint mechanism is effective at reducing interaction risks.

Reviewers found the paper's presentation poor, with unclear paragraphs, inconsistent mathematical notation, and missing details essential for understanding the model structure. Reviewers also noted a lack of discussion on dataset biases and adaptability to diverse clinical contexts. As the AC, I find that medication recommendation tasks formulated solely on ICU settings (MIMIC-III and MIMIC-IV) do not represent a real clinical problem; more experiments on diverse datasets (large-scale, long-term EHR data) are suggested.

**Reviewer Concerns:**

Addressed:

- The authors corrected inconsistent notation, added a notation table, and updated citations regarding VAEs and the "semantic gap" explanation.

- The authors provided a stratified performance analysis based on the number of patient visits to demonstrate effectiveness in cold-start scenarios

- An explanation for the performance drop when removing the DDI loss was provided, clarifying the asymmetric suppression mechanism.





Outstanding:

- Despite revisions, reviewer cR99's fundamental concern regarding the "many details missing to understand the overall structure of the model" remains a major issue.

- Reviewer VBt1 explicitly requested testing on more diverse datasets to address potential biases. The authors did not address this in their response, failing to validate the model on broader, non-ICU datasets, which is critical for this task.

**Reviewer Scores:**

Reviewer cR99 and Reviewer YK7d will likely keep their negative scores. Reviewer VBt1 will possibly lower his/her score upon realizing that their specific concern regarding "biases" and "testing on more diverse datasets" was completely ignored in the authors' response.

---

### Decision · Program_Chairs · 2026-01-26

Reject